

# A decadal time series of water vapor and D/H isotope ratios above Mt. Zugspitze: transport patterns to Central Europe

Petra Hausmann[1], Ralf Sussmann[1], Thomas Trickl[1], and Matthias Schneider[2]

[1]Karlsruhe Institute of Technology, IMK-IFU, Garmisch-Partenkirchen, Germany
[2]Karlsruhe Institute of Technology, IMK-ASF, Karlsruhe, Germany

*Correspondence to*: Ralf Sussmann (ralf.sussmann@kit.edu)

**Abstract.** We present vertical soundings (2005–2015) of tropospheric water vapor ($H_2O$) and its D/H isotope ratio ($\delta D$) at Mt. Zugspitze (47° N, 11° E, 2964 m a.s.l.) derived from ground-based solar Fourier transform infrared (FTIR) measurements. Beside water vapor profiles with optimized vertical resolution (degrees of freedom for signal DOFS = 2.8),
the retrieval provides {$H_2O$, $\delta D$} pairs with consistent vertical resolution (DOFS = 1.6 for $H_2O$ and $\delta D$) applied in this study. The integrated water vapor (IWV) trend of 2.4 [-5.8, 10.6] % decade$^{-1}$ (statistically insignificant, 95 % confidence interval) can be reconciled with the 2005–2015 temperature increase at Mt. Zugspitze (1.3 [0.5, 2.1] K decade$^{-1}$) assuming constant relative humidity. Seasonal variations in free-tropospheric $H_2O$ and $\delta D$ exhibit amplitudes of 140 % and 50 % of the respective overall means. The minima (maxima) in January (July) are in agreement with changing sea surface temperature of
the Atlantic Ocean.

Using extensive backward trajectory analysis, distinct moisture pathways are identified depending on observed $\delta D$ levels: low column-based $\delta D$ values ($\delta D_{col} < 5^{th}$ percentile) are associated with air masses originating in higher latitudes (62° N on average) and altitudes (6.5 km) compared to high $\delta D$ values ($\delta D_{col} > 95^{th}$ percentile: 46° N, 4.6 km). Backward trajectory classification indicates that {$H_2O$, $\delta D$} observations are influenced by three long-range transport patterns towards Mt.
Zugspitze assessed in previous studies: (i) intercontinental transport from North America (TUS; source region: 25–45° N, 70–110° W, 0–2 km altitude), (ii) intercontinental transport from Northern Africa (TNA; source region: 15–30° N, 15° W – 35° E, 0–2 km altitude), and (iii) stratospheric air intrusions (STI; source region: > 20° N, above zonal mean tropopause). The FTIR data exhibit significantly differing signatures in free-tropospheric {$H_2O$, $\delta D$} pairs (given as mean with uncertainty of ± 2 standard errors) for TUS ($VMR_{H_2O}$ = 2.4 [2.3, 2.6] × 10$^3$ ppmv, $\delta D$ = -315 [-326, -303] ‰), TNA (2.8
[2.6, 2.9] × 10$^3$ ppmv, -251 [-257, -246] ‰), and STI (1.2 [1.1, 1.3] × 10$^3$ ppmv, -384 [-397, -372] ‰). For TUS events, {$H_2O$, $\delta D$} observations depend on surface temperature in the source region and the degree of dehydration having occurred during updraft in warm conveyor belts. During TNA events (dry convection of boundary layer air) relatively moist and weakly HDO-depleted air masses are imported. In contrast, STI events are associated to import of predominantly dry and HDO-depleted air masses.
These long-range transport patterns potentially involve the import of various trace constituents to the Central European free troposphere, i.e., import of pollution from North America (e.g., aerosol, ozone, carbon monoxide), Saharan mineral dust,





stratospheric ozone and other airborne species such as pollen. Our above results provide evidence that {H$_2$O, $\delta$D} observations are a valuable proxy for the potential transport of such tracers. To validate this finding, we consult a data base of transport events (TNA and STI) covering 2013–2015 deduced by data filtering from in situ measurements at Mt. Zugspitze and lidar profiles at near-by Garmisch. Indeed, the FTIR data related to these verified TNA events (27 days)

exhibit characteristic fingerprints in IWV (5.5 [4.9, 6.1] mm) and $\delta$D$_{col}$ (-266 [-284, -247] ‰), which are significantly distinguishable from the rest of the time series (4.3 [4.1, 4.5] mm, -316 [-324, -308] ‰). This holds true on 1-$\sigma$ level for 136 STI days (mean ± 1 standard error: 4.2 [4.0, 4.3] mm, -322 [-327, -316] ‰) with respect to the remainder (4.6 [4.5, 4.8] mm, -302 [-307, -297] ‰). Furthermore, deep stratospheric intrusions to the Zugspitze summit (in situ humidity and beryllium-7 data filtering) show a significantly lower mean value (-334 [-337, -330] ‰) of lower-tropospheric $\delta$D (3–5 km a.s.l.) on 2-$\sigma$

level than the rest of the 2005–2015 time series (-284 [-286, -282] ‰). Our results show that consistent {H$_2$O, $\delta$D} observations at Mt. Zugspitze can serve as an operational indicator for long-range transport events potentially affecting regional climate, air quality, as well as human health in Central Europe.

## 1 Introduction

Water vapor (H$_2$O) is of fundamental importance in the climate system of our Earth. As the dominant greenhouse gas it

accounts for about 60 % of the natural greenhouse effect (Kiehl and Trenberth, 1997; Harries et al., 2008). The Earth's energy budget is closely linked to the global water cycle, which effectively redistributes energy by latent heat transport (Stephens et al., 2012; Wild et al., 2013). However, relevant processes determining the interaction between atmospheric humidity, circulation, and climate are still not completely understood (e.g., cloud feedback processes). A major challenge in climate modeling remains to accurately represent the response of hydrological cycle and general circulation patterns to

climate change (Bengtsson et al., 2014; Bony et al., 2015).

Large-scale circulation patterns in the northern midlatitudes are dominated by prevailing westerly winds. Extratropical cyclones and deep convective systems facilitate transport of water vapor from moisture sources to higher altitudes and latitudes. Along these major transport pathways also other trace gas signatures and pollution plumes can travel over long distances from source to receptor regions – possibly even to other continents (e.g., Stohl, 2001; Stohl et al., 2002).

Transported species of predominantly natural origin include, among others, volcanic ash (e.g., Trickl et al., 2013), biomass burning aerosols (e.g., Forster et al., 2001; Wotawa et al., 2001; Damoah et al., 2004; Fromm et al., 2010; Trickl et al., 2015), pollen (e.g., Jochner et al., 2015), mineral dust (e.g., Husar, 2004; Papayannis et al., 2008; Trickl et al., 2011; Israelevich et al., 2012), and stratospheric ozone (e.g., Beekmann et al., 1997; Škerlak et al., 2014). Relevant anthropogenic tracers are, e.g., aerosols, carbon monoxide, ozone, and its precursors. Such transport events impact climate, air quality, and

human health in receptor regions and are highly relevant for agreements on emission regulations (Holloway et al., 2003; TF-HTAP, 2010).





Research on long-range transport to Central Europe has a long history at high-altitude observatories, such as Zugspitze or Jungfraujoch, which offer a unique opportunity to monitor free-tropospheric conditions. Transport studies using lidar and in situ measurements combined with transport modeling identified three main long-range transport patterns to Central Europe. First, intercontinental transport from North America effectively carries anthropogenic emissions within 3–10 days to the

European middle troposphere after lifting by warm conveyor belts (Stohl and Trickl, 1999; Trickl et al., 2003). Second, intercontinental transport from Northern Africa imports Saharan mineral dust into Europe in 5–15 events per year (Papayannis et al., 2008; Flentje et al., 2015). Third, stratosphere–troposphere transport injects dry, ozone-rich air into the European troposphere after typically descending 3–15 days from the lowermost stratosphere (Stohl et al., 2003; Trickl et al., 2010, 2014, 2015, 2016).

Valuable information on tropospheric moisture pathways (and associated transport of other tracers) is provided by measurements of water vapor and its isotopes (Strong et al., 2007; González et al., 2016; Schneider et al., 2016). In this context, two stable water vapor isotopes are of particular interest: $H_2O$ ($^1H_2^{16}O$) and HDO ($^1H^2H^{16}O$) with an average abundance of 99.7 % and 0.03 %, respectively. Variations in the D/H ratio of water vapor are expressed in terms of $\delta D$, i.e., as relative deviation from a reference $HDO$-$H_2O$ ratio standard (Craig, 1961; see Sect. 2). The isotopic composition of

atmospheric water vapor is modified during phase transitions (evaporation, condensation, and sublimation) due to fractionation processes caused by isotopic mass differences (Dansgaard, 1964). After evaporation from the ocean surfaces, which represents the main source of water vapor, air masses are transported to regions of lower temperatures, where condensation or sublimation leads to dehydration and depletion in the heavier isotope HDO. Resulting large-scale isotope effects (Worden et al., 2007; Sutanto et al., 2015) include increasing depletion in HDO (decreasing $\delta D$) with higher latitudes

(latitude effect), with higher altitudes (altitude effect), and with increasing distance from oceans (continental effect).

Only recently, tropospheric water vapor isotope data sets became available from satellite remote sensing instruments (e.g., Worden et al, 2006; Schneider and Hase 2011; Lacour et al., 2012; Boesch et al., 2013; Frankenberg et al., 2013; Sutanto et al., 2015) and from ground-based Fourier transform infrared (FTIR) spectrometers (Barthlott et al., 2016; Schneider et al., 2016) operated within the Network for the Detection of Atmospheric Composition Change (NDACC;

www.ndacc.org). Progress achieved within the project MUSICA (MUlti-platform remote Sensing of Isotopologues for investigating the Cycle of Atmospheric water; Schneider et al., 2012) now offers retrieval methods for tropospheric $H_2O$ and $\delta D$ profiles from mid-infrared FTIR measurements.

So far, research on long-range transport to Central Europe has mainly been based on investigations or field campaigns of special transport events combining observations of conventional tracers (such as ozone, aerosols, and humidity) at few sites.

With the advance of water vapor isotope remote sensing a new promising transport tracer (i.e., consistent {$H_2O$, $\delta D$} pairs) has become available for globally distributed operational FTIR sites. This vast data set can be exploited to gain a more comprehensive picture of atmospheric moisture transport. These transport processes can be associated to the dispersion of anthropogenic (or natural) emissions, which is of particular interest for emission regulation policies and human health issues (e.g., air pollution, pollen distribution). Additionally, analysis of {$H_2O$, $\delta D$} pairs will yield further insight into the coupling



of the hydrological cycle with general circulation patterns, urgently needed to improve our understanding of climate change.

The goal of this study is to evaluate new possibilities in transport research provided by long-term observations of consistent {$H_2O$, $\delta D$} pairs at an FTIR site which is representative for typical midlatitude conditions. We present an update

of the water vapor time series obtained at Mt. Zugspitze (Sussmann et al., 2009) including water vapor isotope information. Based on this data set transport pathways to the Central European free troposphere are identified using backward trajectory analysis. The main task is to identify distinct signatures in {$H_2O$, $\delta D$} pairs for long-range transport patterns and combine the results with conventional transport tracer observations. This paper is structured as follows: Section 2 describes FTIR measurements and the retrieval of water vapor and its isotopes. The resulting long-term {$H_2O$, $\delta D$} time series at Mt.

Zugspitze is presented in Sect. 3 along with a discussion on trends and seasonal cycles. Section 4.1 gives results on moisture pathways to Central Europe related to $\delta D$ outliers. {$H_2O$, $\delta D$} signatures of long-range transport events are identified using trajectory classification (Sect. 4.2) and analysis of lidar and in situ measurements (Sect. 4.3). Finally, Sect. 5 summarizes results and draws final conclusions.

**2 FTIR observations and retrieval strategy**

The long-term time series of water vapor and its isotopes (2005–2015) presented in this study is retrieved from ground-based solar absorption FTIR measurements obtained at Mt. Zugspitze (47.42° N, 10.98° E, 2964 m a.s.l.). This high-altitude observatory is mostly located above the moist boundary layer or just below its upper edge (Carnuth et al., 2002). Sampled air masses are representative of free-tropospheric background conditions over Central Europe. Within the framework of NDACC a Bruker IFS 125HR spectrometer is operated at Mt. Zugspitze (Sussmann and Schäfer, 1997). Mid-infrared solar

absorption spectra are recorded with a typical spectral resolution of 0.005 cm$^{-1}$ (according to a maximum optical path difference of 175 cm) and an integration time of seven minutes (averaging six individual scans). These high-resolution spectra provide information on a large number of trace gases, including water vapor and its isotopes (Sussmann et al., 2009; Schneider et al., 2013).

The isotopic composition of atmospheric water vapor with respect to HDO is typically expressed in terms of $\delta D$, defined

as relative deviation of the HDO-$H_2O$ ratio (volume mixing ratios $VMR_{HDO}$ and $VMR_{H2O}$) from the ratio in Standard Mean Ocean Water ($R_{SMOW} = 3.1152 \times 10^{-4}$; Craig, 1961), i.e.,

$$\delta D = \left( \frac{VMR_{HDO}/VMR_{H2O}}{R_{SMOW}} - 1 \right) \times 1000\ ‰ . \tag{1}$$

Additionally, columnar water vapor and $\delta D$ values are applied in the following: column-based $\delta D$ ($\delta D_{col}$) is calculated using the total column ratio in Eq. (1) and integrated water vapor (IWV) is thereafter given in units of mm (i.e., a water column of

1 mm corresponds to an atmospheric column density of $3.345 \times 10^{+21}$ molecules cm$^{-2}$, which can be derived considering the Avogadro constant and the molar mass as well as the density of water).



The retrieval of water vapor isotopes from mid-infrared FTIR spectra is very demanding due to the high variability of atmospheric water vapor compared with relatively small $\delta$D variations that result from the strong correlation of HDO and $H_2O$ abundances. Retrieval strategies for tropospheric {$H_2O$, $\delta$D} pairs from ground-based FTIR spectra were developed within the project MUSICA (Schneider et al., 2012, 2016; Barthlott et al., 2016). The latest retrieval version (v2015) is

applied here, which comprises − briefly summarized − a logarithmic-scale retrieval with an interspecies constraint between HDO and $H_2O$, a non-Voigt line-shape model, and a priori profiles from the LMDZ model. Two data products are available: (i) optimal estimation of water vapor for maximum vertical resolution of $H_2O$ profiles (type 1) and (ii) quasi optimal estimation of {$H_2O$, $\delta$D} pairs derived by a posteriori data processing for consistent vertical sensitivity of $H_2O$ and $\delta$D profiles (type 2). Error estimation for the type 2 product reveals a precision below 2 % for integrated water vapor and below

30 ‰ for column-based $\delta$D (Schneider et al., 2012).

Applying this {$H_2O$, $\delta$D} pair retrieval to Zugspitze FTIR spectra results in characteristic averaging kernels depicted in Fig. 1 for the humidity-proxy state, i.e., $\frac{1}{3} \times (\ln[H_2^{16}O] + \ln[H_2^{18}O] + \ln[HD^{16}O])$. Furthermore, the vertical sensitivity is shown, which is defined as sum of the row elements of the averaging kernel matrix. An exemplary measurement on 29 October 2009 is chosen to represent the mean state of the Zugspitze time series (IWV = 3.8 mm, degrees of freedom for

signal DOFS = 1.6 for the type 2 product). Optimally estimated $H_2O$ profiles (type 1 product) derived from Zugspitze data provide an average DOFS of 2.8 (standard deviation SD = 0.2) and are sensitive up to 12 km altitude (vertical sensitivity larger than 0.75) as shown in Fig. 1a. In the following, we use the type 2 data product (Fig. 1b), which provides consistent {$H_2O$, $\delta$D} profiles with vertical sensitivity up to 9 km (SD = 1 km) and maximum sensitivity in 5 km a.s.l. altitude (SD = 0.5 km). On average, a DOFS of 1.6 (SD = 0.2) is obtained for $H_2O$ and $\delta$D data (type 2) derived from Zugspitze spectra.

The vertical resolution amounts to 2–3 km in the lower troposphere and 4–5 km in the middle-upper troposphere, as derived from the full width at half maximum of the row kernels. Retrieval quality selection is implemented by means of a threshold in the root-mean-square residuals of the spectral fit (RMS), which is chosen to eliminate 5 % of all spectra with the highest RMS (see Sussmann et al., 2009). The full Zugspitze time series yields a mean normalized RMS of 0.26 %. Additionally, the sum of DOFS for all retrieved isotopes ($H_2O$, HDO, and $H_2^{18}O$) is required to exceed a value of 4.0. To compute daily

means, only days with more than one measurement are considered.

## 3 Long-term {$H_2O$, $\delta$D} time series and its seasonality above Mt. Zugspitze

Time series of daily mean integrated water vapor and column-based $\delta$D above Mt. Zugspitze are presented in Fig. 2 for the period 2005–2015. This data set comprises 1154 daily means derived from 10184 quality selected FTIR spectra with, on average, nine spectra per measurement day and 105 measurement days per year. Seasonal cycles are determined by fitting a

third-order Fourier series to the time series. This fitted intra-annual model is subtracted from the time series for the purpose of deseasonalization. The multiannual mean of the deseasonalized IWV time series amounts to 4.4 ± 0.1 mm (mean ± 2 SE, i.e., standard error of the mean SE = SD/$\sqrt{n}$ with sample size $n$), which reflects dry conditions at the high-altitude Zugspitze





site.). The multiannual mean of deseasonalized column-based $\delta D$ amounts to -311.3 ± 3.8 ‰. This relatively low value corresponds to strong HDO depletion at high altitudes according to the altitude effect (dehydration with decreasing temperature). The multiannual mean $\delta D_{col}$ value at Zugspitze is slightly less depleted than observations at nearby, but even higher Jungfraujoch station (46.5° N, 8.0° E, 3580 m a.s.l.) with a multiannual mean $\delta D_{col}$ of -330 ‰ (Schneider et al.,

5   2012).

A short discussion is given on long-term trends derived from the daily mean IWV and $\delta D_{col}$ time series (2005–2015). For column-based $\delta D$, we infer a statistically insignificant trend of 0.8 [-3.1, 4.7] % per decade (relative to the overall mean, uncertainty given as 95 % confidence interval). For IWV, a weak positive, also insignificant trend of 2.4 [-5.8, 10.6] % per decade (relative to overall IWV mean) is derived. These linear trends are determined by fitting a linear trend model by least

square fit to the deseasonalized daily mean time series and uncertainties are determined via bootstrap resampling of the residuals (Gardiner et al., 2008). Over the same time period, a significant positive temperature trend of 1.3 [0.5, 2.1] K per decade is observed at Mt. Zugspitze (derived from daily means of in situ temperature measurements coincident with FTIR spectra within a period of ± 30 min). Assuming constant relative humidity (RH) and following the Clausius–Clapeyron equation (Schneider et al., 2010), the observed temperature increase translates to a positive IWV trend of 9.2 [3.7, 14.7] %

per decade. This calculated trend agrees with the observed IWV trend within uncertainties. The assumption of constant RH is valid on large spatial scales, but RH slightly decreased over specific land regions (Hartmann et al., 2013), which would explain the relatively large calculated IWV trend compared to observations.

Strong seasonal cycles of free-tropospheric water vapor and $\delta D$ are observed above Zugspitze (Fig. 3a). Multiannual monthly means are shown for data at 5 km a.s.l., where the maximum in vertical sensitivity of the $\{H_2O, \delta D\}$ pair product is

located (see Fig. 1b). These data are representative for an altitude region of 3–7 km, which is the full width at half maximum of the corresponding averaging kernel row. The seasonal cycle amplitude (relative difference of maximum and minimum monthly mean) amounts to 140 % for $H_2O$ and 50 % for $\delta D$ relative to the overall mean. Both seasonal cycles show maxima in summer (July) and minima in winter (January). The $\delta D$ maximum in summer results from more frequent ascending motions in summer with associated transport of less HDO-depleted air masses from lower altitudes and latitudes (Risi et al.,

2012). Stronger HDO depletion in winter ($\delta D$ minimum) is caused by a stronger continental temperature gradient in winter and associated stronger dehydration during air mass transport from the Atlantic Ocean. Additionally, transport from Eastern Europe occurs more frequently during anticyclonic conditions in winter importing strongly depleted continental air masses (Christner, 2015). Monthly frequency distributions of single $H_2O$ measurements are generally right-skewed and variability is larger in summer (June–August) than in winter. This might be explained by extreme moistening events due to more frequent

convection in summer and mixing with boundary layer air. For free-tropospheric $\delta D$ measurements the monthly frequency distributions are moderately left-skewed, which points to episodic influence by strongly HDO-depleted air masses (e.g., originating in the upper troposphere or lower stratosphere). In the $\{H_2O, \delta D\}$ distribution plot (Fig. 3b), it becomes clear that the isotopic composition of atmospheric water vapor significantly changes over the course of a year: in spring it is less depleted in HDO than in autumn. This is mainly caused by seasonal variations in sea surface temperature of the Atlantic





Ocean (i.e., the major moisture source) as derived from Rayleigh model simulations (see Fig. 5). An additional contribution is possibly due to increased mixing in spring compared to autumn. These seasonal variations in the water vapor isotopic composition above Zugspitze imply that $\delta$D observations provide information complementary to the $H_2O$ data (e.g., Risi et al., 2012; Schneider et al., 2012).

In this section, we presented a unique decadal time series of water vapor and $\delta$D derived from FTIR measurements at Mt. Zugspitze representative of Central European background conditions. In the following, these $\delta$D observations are used as a tracer of atmospheric transport to study its behavior in relation to different transport pathways to Central Europe.

## 4 Transport patterns to the Central European free troposphere

### 4.1 Moisture pathways related to $\delta$D outliers

To explore the potential of water vapor isotope observations as a proxy for transport processes, moisture transport pathways are identified for cases of extreme water vapor isotopic composition. The Zugspitze data set of {$H_2O$, $\delta$D} pairs contains predominantly information on free-tropospheric moisture pathways reaching Central Europe in the altitude region around 5 km a.s.l., as shown in Sect. 2 (see Fig. 1b). With the help of backward trajectory analysis, these transport pathways are determined for outliers in column-based $\delta$D derived from the frequency distribution of the deseasonalized daily mean time

series at Mt. Zugspitze (2005–2015). High outliers are defined as daily mean values larger than 95[th] percentile of the distribution ($\delta D_{col}$ > -219 ‰) and low outliers are days with $\delta D_{col}$ below the 5[th] percentile ($\delta D_{col}$ < -430 ‰). This yields 56 days of high and 57 days of low column-based $\delta$D outliers from a total of 1154 measurement days.

For all 2005–2015 FTIR measurements 120 hour backward trajectories arriving in 5 km a.s.l. above Zugspitze are calculated using the Air Resources Laboratory's HYbrid Single-Particle Lagrangian Integrated Trajectory model (HYSPLIT;

available at http://ready.arl.noaa.gov; Stein et al., 2015). Based on positive experience, we apply meteorological data from the NCEP reanalysis (global 2.5° grid) and model vertical velocity. Trajectory uncertainty is estimated to 10–20 % of the travel distance (Stohl, 1998), which is fulfilled in the free troposphere according to our experience. Resulting backward trajectories are shown in Fig. 4 as horizontal and vertical projections. The initial point of the trajectories is chosen as the point of last condensation (LC), defined as region where the relative humidity along the trajectory exceeds 80 % over a three

hour period (see González et al., 2016). If no condensation occurred (i.e., no LC point found), full 120 hour trajectories are depicted. As water vapor mixing ratio and isotopic composition are conserved in absence of sources and sinks (Galewsky et al., 2005; Noone et al., 2012), conditions at the LC point determine observed {$H_2O$, $\delta$D} pairs if mixing during transport is negligible.

According to the location of Mt. Zugspitze in the zone of prevailing westerlies and in agreement with results from long-

term ozone lidar measurements at nearby Garmisch site (see Sect. 4.3), the majority of trajectories point to Atlantic moisture sources distributed over a wide latitude range (from subtropics to polar regions). Several trajectories originate over the North American continent or even the Pacific region. Fewer trajectories arrive from Eastern Europe or Northern Africa. Figure 4





reveals clearly different transport patterns for low outliers of column-based $\delta$D observed at Zugspitze compared to high outliers. For extraordinarily low $\delta D_{col}$ observations air masses mostly descend from high latitudes and altitudes, often related to cold advection at the rear side of an upper level trough. In contrast, on days with very high $\delta D_{col}$ observations, air masses predominately arrive from lower latitudes ascending from lower altitudes, often in connection with warm advection at the

front side of an upper level trough. An overview of the conditions at the last condensation points and their positions is given in Table 1 for trajectories arriving on $\delta$D outlier days (only if LC point is found along the trajectory). This analysis shows that air masses originate from significantly higher latitudes (about 60° N) and altitudes (about 6.5 km) on low-$\delta$D outlier days than on high-$\delta$D days. Last condensation occurred at significantly lower temperatures and under dryer conditions for low-$\delta$D days compared to high-$\delta$D days. Consequently, mostly dry air masses are transported to Zugspitze in connection

with low $\delta$D.

A first order interpretation of the {$H_2O$, $\delta$D} data pairs can be obtained by their comparison to theoretical Rayleigh and mixing lines (Wiegele et al., 2014; Schneider et al., 2015; González et al., 2016). Figure 5 shows the {$H_2O$, $\delta$D} distribution plot for Zugspitze data at 5 km a.s.l. along with simulated Rayleigh and mixing processes. Rayleigh models simulate the idealized process of gradual dehydration of an air parcel during adiabatic cooling with immediate removal of the condensate.

Simultaneously, the remaining water vapor becomes more and more depleted in HDO, as heavier isotopes preferentially condense. Rayleigh dehydration processes (black lines in Fig. 5) are simulated using a mean midlatitude profile of pressure and temperature (Christner, 2015, Table 13) with initial evaporation conditions characteristic of midlatitude oceanic or continental moisture sources (temperature of 15 °C, RH of 80 %, and different $\delta$D values: -60 ‰, -130 ‰, and -160 ‰). Furthermore, mixing processes of dry, HDO-depleted upper-tropospheric air ($VMR_{H2O}$ = 200 ppmv, $\delta$D = -585 ‰) with

moist lower-middle-tropospheric air masses are simulated (dark red lines in Fig. 5). The $\delta$D value of the mixture is mainly determined by the $\delta$D value of the mixing partner with higher water vapor (Noone et al., 2011). Three moist mixing partners are considered here: (i) boundary layer air with $VMR_{H2O}$ = 13500 ppmv and $\delta$D = -130 ‰, (ii) moderately dehydrated and depleted air with $VMR_{H2O}$ = 6100 ppmv and $\delta$D = -200 ‰, and (iii) even more dehydrated air with $VMR_{H2O}$ = 3000 ppmv and $\delta$D = -270 ‰. Comparing these simulated transport processes with {$H_2O$, $\delta$D} pairs observed at Zugspitze (5 km a.s.l.)

reveals underlying transport regimes on designated $\delta D_{col}$ outlier days: measurements on days of low-$\delta$D outliers group along Rayleigh curves, while data on high-$\delta$D outlier days preferentially group along simulated mixing lines. We tentatively conclude that the first group (low $\delta D_{col}$) experienced slow ascent from mostly midlatitude moisture sources with associated gradual dehydration by condensation and rainout, which is followed by subsidence to the middle troposphere above Zugspitze with only minor mixing with moist and less HDO-depleted air masses. In contrast, the second group (high $\delta D_{col}$)

reveals air masses ascending from lower altitudes and latitudes, which are influenced by co-existence or mixing of moist lower-tropospheric and drier middle-upper-tropospheric air masses.

The presented $\delta$D-outlier analysis reveals the valuable potential of water vapor isotope observations for investigating moisture transport pathways. As along these pathways also many other atmospheric species can be transported, {$H_2O$, $\delta$D} data pairs might also serve as useful tracer in long-range transport research.



## 4.2 {H₂O, δD} signatures of long-range transport events

Moisture transport pathways to the Central European free troposphere were identified for days with extraordinary high or low $\delta D$ observations at Mt. Zugspitze in Sect. 4.1. Going a step further, we examine to what extent {H₂O, $\delta D$} observations provide information also on long-range transport events of various other atmospheric tracers. In the following, {H₂O, $\delta D$} signatures are analyzed for three FTIR measurement categories, each of which is influenced by a distinct long-range transport pattern to the Central European free troposphere: (i) intercontinental transport from North America (TUS), (ii) intercontinental transport from Northern Africa (TNA), and (iii) stratospheric air intrusions (STI). Zugspitze FTIR measurements are assigned to these categories by classification of respective backward trajectories (see Sect. 4.1) using the criteria given in Table 2 and described in more detail in the following paragraphs.

Intercontinental transport from North America (TUS) may effectively carry anthropogenic pollution plumes (e.g., ozone, aerosols) to Central Europe within typically 3−10 days (Stohl and Trickl, 1999; Trickl et al., 2003; Huntrieser et al., 2005). Consequently, North American emissions strongly contribute to the European total column tracer mass (i.e., 43 % for a tracer with 10 days lifetime; Stohl et al., 2002). The typical pathway of polluted boundary layer air from North America to Europe is uplift in a warm conveyor belt (WCB; i.e., an ascending air stream ahead of a surface cold front in an extratropical cyclone) and subsequent transport by strong westerly flows in the middle and upper troposphere. The North American tracer enters Europe typically at altitudes of 4–8 km and at high latitudes (> 60° N). Here, the circulation frequently turns anticyclonic and tracers eventually reach Northern Alps (Huntrieser and Schlager, 2004). WCB climatologies reveal a major inflow region at the southeastern coast of North America (Stohl et al., 2001; Eckhardt et al., 2004; Madonna et al., 2014). In the following, Zugspitze backward trajectories passing this source region (25–45° N, 70–110° W, 0–2 km altitude) are assigned to the TUS category. The second transport category considered is intercontinental transport from Northern Africa (TNA) to the European free troposphere. Associated transport of Saharan mineral dust influences air quality, soil fertility, radiative budget, and atmospheric oxidation capacity in the receptor regions (Ravishankara, 1997). Each year 5−15 events of mineral dust import occur in Central Europe each lasting 1−3 days (Papayannis et al., 2008; Flentje et al., 2015). The TNA transport category includes backward trajectories passing the Saharan boundary layer region (15−30° N, 15° W − 35° E, 0–2 km altitude; Engelstaedter et al., 2006). The third transport class accounts for extratropical stratospheric intrusions (STI), which occur mainly in synoptic-scale processes such as tropopause folds and cutoff lows near the polar or subtropical jet stream (Stohl et al., 2003). Filaments of ozone-rich stratospheric air descend from the lowermost stratosphere and proceed to the Central European free troposphere via several pathways (Trickl et al., 2010, 2011; Škerlak et al. 2014). Mixing with tropospheric air might exhibit relatively long time scales as little modification is reported even after several days of transport (Trickl et al., 2014). In the following, Zugspitze backward trajectories originating above the zonal mean tropopause (TP) taken from ECMWF data (Eckhardt et al., 2004) at latitudes above 20° N are assigned to the STI category (required minimum residence time of five hours above TP, which is penetrated at least once by more than 1 km). This TP definition is chosen, as potential vorticity for a dynamical TP definition is not provided within HYSPLIT.



These long-range transport categories (TUS, TNA, and STI) are expected to have characteristic imprints on {$H_2O$, $\delta D$} pairs observed at Mt. Zugspitze. Stratospheric intrusions originate in the lowest few kilometers of the stratosphere (Trickl et al., 2014, 2016), where moisture content is extremely low and the mean $\delta D$ profile exhibits a minimum before increasing above due to growing influence of isotopically heavier water vapor from methane oxidation (Zahn et al., 2006; Steinwagner

et al., 2010). Consequently, STI events potentially import relatively dry and HDO-depleted air masses to the Central European troposphere. By contrast, TUS and TNA air masses originate in the moist boundary layer and may transport relatively moist and less HDO-depleted air masses to Central Europe. However, strong WCB updraft during TUS events may cause air mass dehydration and HDO depletion through rainout (Rayleigh process).

The resulting Zugspitze backward trajectory categories obtained by the transport criteria described above are presented in

Fig. 6. Distributions of deseasonalized $VMR_{H2O}$ and $\delta D$ for the corresponding classes of free-tropospheric Zugspitze FTIR data (5 km a.s.l.) are depicted in Fig. 7. Considering uncertainty of two standard errors ($\pm$ 2 SE), mean $VMR_{H2O}$ and $\delta D$ values are significantly different for all three transport classes (see Table 2). As expected, STI is associated with the driest and most HDO-depleted air masses. TNA is connected to moister and less depleted air, while TUS exhibits intermediate $VMR_{H2O}$ and $\delta D$ values. The standard deviation of $VMR_{H2O}$ distributions is similar for all transport classes (SD $\sim 10^3$ ppmv).

In the case of $\delta D$ distributions, TNA shows less scatter (SD = 34 ‰) compared to TUS (SD = 65 ‰) and STI (SD = 73 ‰). This indicates a quite homogeneous source region and transport regime for TNA. Larger scatter for TUS might result from variable strengths of WCB uplift causing various levels of dehydration and HDO depletion. Relatively large scatter in the $\delta D$ distribution for STI events implies that also air masses weakly depleted in HDO are observed during STI events. This could have several reasons: First, stratospheric intrusions with depths ranging from few hundreds to several thousands of meters

are not necessarily resolved by the relatively coarse vertical resolution of the FTIR $\delta D$ product. Second, as a climatological tropopause altitude is applied to identify STI events, local and seasonal TP variations are not accounted for, i.e., also (upper-) tropospheric trajectories might be included in the STI class. Third, mixing with tropospheric air might occur during transport from the stratosphere, which was found, however, to be very slow in the free troposphere (Trickl et al., 2014). Fourth, $\delta D$ values in the lower stratosphere might be less depleted than predicted from Rayleigh processes, which is probably caused by

extratropical troposphere-stratosphere transport, e.g., by convectively lofted ice (Hanisco et al., 2007; Steinwagner et al., 2010; Randel et al., 2012). All these mechanisms would yield higher $\delta D$ observations at Zugspitze than expected for STI events from the $\delta D$ minimum in the lower stratosphere.

In analogy to Fig. 5, {$H_2O$, $\delta D$} data pairs for these long-range transport patterns are interpreted by comparison to theoretical Rayleigh and mixing lines as depicted in Fig. 8. For TNA events, we derive from this analysis that moist

boundary layer air is imported to the free troposphere by means of dry convection without significant condensation and cloud formation (González et al., 2016; Schneider et al, 2016). Therefore, TNA air masses are less depleted in HDO (higher $\delta D$ values). In case of TUS events, either dry or moist air masses can be imported to Central Europe. TUS transport is generally in line with simulated Rayleigh processes and suggests that moist TUS air masses originate over warmer surfaces while drier TUS air masses have their origin over colder regions. The STI transport category also largely agrees with





Rayleigh models. In case of dry STI events, air masses seem to be partially influenced by modeled mixing processes of dry upper-tropospheric or lower-stratospheric air with free-tropospheric air masses.

Overall, we find distinct {$H_2O$, $\delta D$} fingerprints in Zugspitze FTIR data for three categories of long-range transport patterns (TUS, TNA, and STI). Consequently, consistent {$H_2O$, $\delta D$} observations can serve as a valuable new transport

tracer to study atmospheric transport to the Central European free troposphere, especially if combined with measurements of conventional transport tracers.

## 4.3 Combination with lidar and in situ measurements of transport tracers

Transport categories analyzed in Sect. 4.2 identify backward trajectories and FTIR measurements which are potentially influenced by long-range transport of relevant atmospheric tracers to Central Europe. However, this analysis cannot

determine whether tracers other than {$H_2O$, $\delta D$} were actually transported to a significant extent along the trajectory to the receptor region. Actual tracer transport would require substantial emissions in the respective source region during air mass overpass and depends on tracer reactivity as well as on meteorological conditions during transport to Central Europe. In the following, we use lidar and in situ observations of conventional transport tracers (i.e., ozone, aerosols, and humidity) to identify events of long-range tracer transport reaching the Northern Alps and to analyze the respective {$H_2O$, $\delta D$} signatures

in Zugspitze FTIR measurements.

Ozone and aerosol lidar measurements obtained at Garmisch site (47.48° N, 11.06° E, 743 m a.s.l.) close to Mt. Zugspitze offer the possibility to detect long-range tracer transport events reaching Northern Alps. We analyze Garmisch lidar observations in the period 2013–2015 to identify mineral dust import from Northern Africa and stratospheric air intrusions. Mineral dust transport from Northern Africa (TNA) becomes apparent in lidar profiles as elevated aerosol layers,

which are attributed to Northern African origin by HYSPLIT trajectory analysis. Stratospheric intrusions (STI) appear in lidar profiles as layers with a vertical extent from 200 meters up to several kilometers, which exhibit ozone enhancements of the order of 10 ppb or more relative to ozone concentrations observed in the layers above and below. Corresponding layers of low relative humidity (RH < 10 %) are found in radiosonde profiles recorded in Munich (48.25° N, 11.55° E, 492 m a.s.l.) or in water vapor profiles from differential absorption lidar measurements at Mt. Zugspitze (Vogelmann et al., 2015). These

dry, ozone-rich layers are assigned to being of stratospheric origin either by 4day stratospheric intrusion forecasts or by 315h HYSPLIT trajectories (Trickl et al., 2010).

In the years 2013–2015, Garmisch lidar observations reveal 41 days influenced by TNA events and 242 days of STI events from a total of 360 observation days. Coincident FTIR measurements at Mt. Zugspitze are available on 27 (136) days of these TNA (STI) days, which corresponds to 9 % (47 %) of all FTIR measurement days in the 2013–2015 period. Figure 9

and Table 3 present distributions of deseasonalized daily mean IWV and $\delta D_{col}$ for this compilation of FTIR measurements. For TNA days with mineral dust transport we find significantly higher mean IWV and less HDO depletion (IWV = 5.47 [4.90, 6.05] mm, $\delta D_{col}$ = -266 [-284, -247] ‰) compared to all FTIR data in 2013–2015 excluding the TNA events (significance on 2-σ level). This result can be related to transport of less depleted air masses from lower altitudes and





latitudes during TNA events. Additionally, mineral dust transport requires that no condensation (i.e., dehydration and depletion) occurred along the pathway to avoid wet deposition of aerosols. In contrast, STI events show a tendency to lower IWV and stronger HDO depletion (IWV = 4.18 [4.01, 4.34] mm, $\delta D_{col}$ = -322 [-327, -316] ‰) compared to all FTIR data excluding STI events (significant difference on 1-σ level, i.e., mean ± 1 SE, see Table 3). The broad $\delta D$ distribution for STI days is similar to results found in Sect. 4.2 and can probably be explained by similar arguments.

Frequently, stratospheric intrusions penetrate deep into the troposphere, eventually reaching mountain summit observatories or even surface stations (e.g., Eisele et al., 1999; Schuepbach et al., 1999; Stohl et al., 2000; Lefohn et al., 2012; Lin et al., 2012; Itoh and Narazaki, 2016; Ott et al., 2016). In the following, deep stratospheric intrusions (DSTI) are defined as STI events reaching the summit of Mt. Zugspitze. DSTI events can be detected by in situ measurements of stratospheric tracers, such as high beryllium-7 (Be-7) and low relative humidity (Trickl et al., 2010). The cosmogenic radionuclide beryllium-7 is a good, but not unambiguous indicator for stratospheric air, as it is produced mainly (i.e., to 67 %; Lal and Peters, 1967) in the stratosphere. Following previous studies (e.g., Trickl et al., 2010), deep stratospheric intrusions to the Zugspitze summit can be identified by combinations of tracer thresholds: (i) RH < 60 % and Be-7 larger than the 85[th] percentile of the annual Be-7 distribution (flag 1) and (ii) RH < 60 % combined with RH < 30 % for at least one data point within six hours before and after the respective measurement (flag 2). For the period 2005–2015, in situ measurements at Mt. Zugspitze provide 12h averages of Be-7 and hourly RH data (T. Steinkopff, pers. comm., 2016), which are displayed in Fig. 10. Deep stratospheric intrusion periods are marked as identified using flag 1. In addition, Fig. 10 shows coincident FTIR observations of lower-tropospheric $\delta D$ (deseasonalized hourly means of partial columns from 3–5 km a.s.l.). From a total of 6560 coincident hourly means in 2005–2015, 24 % are identified as DSTI events using in situ flag 1 and 38 % if using flag 2. The tendency of flag 2 to cover more intrusion cases is expected from results in Trickl et al. (2010).

The distribution of lower-tropospheric $\delta D$ for deep stratospheric intrusion events is depicted in Fig. 11. The mean value of lower-tropospheric $\delta D$ is significantly lower for DSTI events than for the full time series without these events. It amounts to -334 [-337, -330] ‰ (mean ± 2 SE) for DSTI-flag 1 and, for flag 2, it amounts to -331 [-334, -328] ‰. For the full time series excluding DSTI events from flag 1 (flag 2), the mean value is -284 [-286, -282] ‰ (-274 [-276, -273] ‰). The mean of the full time series amounts to -296 [-298, -294] ‰. Scatter of lower-tropospheric $\delta D$ distributions is about 70 ‰ for both DSTI flags. Overall, both flags for the identification of deep stratospheric intrusions from in situ observations provide consistent results with respect to the distribution of lower-tropospheric $\delta D$ derived from coincident FTIR measurements. The significant shift of the lower-tropospheric $\delta D$ distribution to lower $\delta D$ values for DSTI events at Mt. Zugspitze agrees with the conception of intrusions originating in the strongly HDO-depleted lower stratosphere region. Nevertheless, the $\delta D$ distribution for DSTI events exhibits a relatively large scatter, which implies that also weakly depleted air masses are found for DSTI events. This result is in line with relatively broad $\delta D$ distributions found for the STI transport category in Sect. 4.2 and for STI events identified from lidar measurements (see Fig. 9).





Correlation analysis for all coincident measurements of lower-tropospheric $\delta$D (FTIR) and Be-7 (in situ) yields a significant negative correlation (99 % confidence) with a correlation coefficient of $R$ = -0.295. This negative correlation is in line with expectations, as Be-7 concentrations increase with altitude towards the lower stratosphere, whereas $\delta$D values decrease with altitude.

**5 Summary and conclusions**

In this study, we presented a decadal time series of water vapor and $\delta$D above Mt. Zugspitze (2005−2015) derived from mid-infrared FTIR measurements within the NDACC framework, which are representative of Central European background conditions. The Zugspitze {$H_2O$, $\delta$D} data product provides an average DOFS of 1.6 for consistent $H_2O$ and $\delta$D profiles with a maximum vertical sensitivity in the free troposphere (around 5 km a.s.l.). For the time period 2005−2015, we found no

statistically significant trend of integrated water vapor and column-based $\delta$D. However, the weakly positive (insignificant) IWV trend can be reconciled with the significant temperature increase at Zugspitze over this time period assuming constant relative humidity. Both, IWV and $\delta D_{col}$, exhibit strong seasonal cycles with summer maxima and winter minima. In the {$H_2O$, $\delta$D} distribution plot, it becomes obvious that water vapor isotope data provide additional information not contained in $H_2O$ observations alone.

Consistent {$H_2O$, $\delta$D} observations are a valuable proxy for atmospheric transport pathways. We demonstrated the potential of this new transport tracer by analyzing a vast compilation of backward trajectories to the free troposphere above Mt. Zugspitze (5 km a.s.l.). Distinct moisture pathways to the Central European free troposphere were identified for days of extraordinarily high or low column-based $\delta$D observations at Mt. Zugspitze. While low-$\delta$D observations are predominantly associated with descending dry air masses from higher latitudes, high-$\delta$D data are mostly related to ascending moist air

masses from lower latitudes.

Significantly different {$H_2O$, $\delta$D} signatures were found for three main long-range transport pathways to Central Europe, i.e., stratospheric intrusions (STI), intercontinental transport from North America (TUS), and from Northern Africa (TNA). We identified Zugspitze FTIR measurements influenced by these transport patterns by means of backward trajectory classification. The corresponding free-tropospheric $VMR_{H2O}$ distributions exhibit mean values (± 2 SE) of 1.2 [1.1, 1.3] ×

$10^3$ ppmv (STI), 2.4 [2.2, 2.6] × $10^3$ ppmv (TUS), and 2.8 [2.6, 2.9] × $10^3$ ppmv (TNA). Mean free-tropospheric $\delta$D values are -384 [-397, -372] ‰ (STI), -315 [-326, -303] ‰ (TUS), and -251 [-257, -246] ‰ (TNA). The finding of distinct {$H_2O$, $\delta$D} fingerprints was validated for verified STI and TNA events, deduced by data filtering from 2013−2015 lidar and in situ measurements at Mt. Zugspitze and near-by Garmisch. Mean values of IWV and $\delta D_{col}$ for these TNA (STI) events are significantly different on 2-σ level (1-σ level) with respect to the rest of the time series. Also deep stratospheric intrusions to

the Zugspitze summit (data filtering of 2005−2015 in situ humidity and beryllium-7 data) reveal a significantly lower mean value of lower-tropospheric $\delta$D (3−5 km a.s.l.) than the full time series on 2-σ level.





These results confirm import of dry and strongly HDO-depleted air masses expected for STI events, which originate in the extremely dry and HDO-depleted lowermost stratosphere. STI events generally agree with Rayleigh simulations, but may be influenced by mixing processes of dry lower-stratospheric air with free-tropospheric air masses. In contrast, TNA air masses originate in the moist boundary layer and are associated with dry convection and import of relatively moist and less HDO-depleted air masses to Central Europe. Also TUS air masses have their origin in the boundary layer and their humidity depends on the surface temperature of their oceanic or continental source region. However, WCB updrafts during TUS events may cause variable degrees of air mass dehydration and HDO depletion. Overall, $\{H_2O, \delta D\}$ observations add valuable new information to previous long-range transport studies using lidar and in situ measurements of conventional transport tracers (i.e., humidity, ozone, beryllium-7, and aerosols), which offer the possibility to identify transport events of relevant tracers actually reaching Northern Alpine stations.

In future work, backward trajectory classification to identify STI events could be improved, e.g., by application of a dynamical tropopause definition. $\{H_2O, \delta D\}$ observations could also be combined with FTIR data of other relevant trace species, such as ozone or carbon monoxide, for more reliable detection of transport events. Furthermore, $\{H_2O, \delta D\}$ profile information provided by FTIR measurements (approximately two tropospheric partial columns) could be combined with lidar or radiosonde humidity profiles with high vertical resolution, especially for studying thin filaments of stratospheric air intrusions. Regarding long-term trends, water vapor trends could be estimated for different transport categories to investigate to what extent changing transport patterns contribute to observed water vapor trends. Of particular importance would be research on the connection between lower-stratospheric humidity trends, trends in the frequency of stratosphere–troposphere transport events, and tropospheric water vapor trends.

Finally, combining $\{H_2O, \delta D\}$ pair measurements from all globally distributed NDACC sites with observations from in situ and lidar networks and operational transport modeling could provide a unique new database to study the impact of atmospheric transport with respect to climate, air quality, and human health. $\{H_2O, \delta D\}$ data could serve as an indicator for the presence of pronounced transport events at Mt. Zugspitze to initiate case studies using observations of the respectively relevant tracers (e.g., humidity, aerosols, ozone, pollen). On demand, further FTIR sites could be included, which are located along simulated transport trajectories. Particular focus has to be laid on deepening our knowledge on the coupling of changing transport patterns in the global circulation, changes in the intensity of the hydrological cycle, and climate change.

**Acknowledgments**

We thank H. P. Schmid (IMK-IFU) for his continual interest in this work. We gratefully acknowledge Frank Hase (IMK-ASF) for his support in using PROFFIT and the NOAA Air Resources Laboratory for the providing the HYSPLIT transport and dispersion model (www.ready.noaa.gov). Our work has been funded by the Bavarian State Ministry of the Environment and Consumer Protection via grant VAO-II TPI/01. We thank for support by the Deutsche Forschungsgemeinschaft and



Open Access Publishing Fund of the Karlsruhe Institute of Technology. This study has benefit from progress achieved in the framework of the European Research Council project MUSICA (FP7/(2007-2013)/ERC grant agreement number 256961).

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





**Table 1.** Air mass origin and conditions at last condensation point for trajectories on $\delta$D outlier days (only if LC is found) given as mean values with uncertainties of two standard errors (SE) and range of minimum and maximum values.

| | High $\delta D_{col}$ ($> 95^{th}$ perc.) | | Low $\delta D_{col}$ ($< 5^{th}$ perc.) | |
| --- | --- | --- | --- | --- |
| | Mean (± 2 SE) | Min, Max | Mean (± 2 SE) | Min, Max |
| **Latitude (° N)** | 46 [45, 47] | 30, 55 | 62 [61, 63] | 34, 75 |
| **Altitude (km a.s.l.)** | 4.6 [4.4, 4.9] | 0.4, 8.1 | 6.5 [6.4, 6.6] | 3.3, 8.7 |
| **Pressure (hPa)** | 579 [560, 599] | 333, 963 | 438 [432, 445] | 317, 678 |
| **Temperature (K)** | 260 [259, 262] | 226, 289 | 242 [241, 243] | 224, 274 |
| **$VMR_{H2O}$ ($10^3$ ppmv)** | 4.1 [3.7, 4.5] | 0.3, 15.6 | 1.2 [1.1, 1.3] | 0.2, 8.5 |



**Table 2.** Backward trajectory classification by source region and resulting distributions of deseasonalized $VMR_{H2O}$ and $\delta D$ for the long-range transport patterns of stratospheric intrusions (STI), transport from North America (TUS), and Northern Africa (TNA).

| | STI | TUS | TNA |
|---|---|---|---|
| **Trajectory source region:** | | | |
| **Latitude** | > 20° N | 25–45° N | 15–30° N |
| **Longitude** | – | 70–110° W | 15° W–35° E |
| **Altitude** | > zonal mean TP | 0–2 km | 0–2 km |
| **{$H_2O$, $\delta D$} signature (mean ± 2 SE):** | | | |
| **$VMR_{H2O}$ ($10^3$ ppmv)** | 1.21 [1.07, 1.34] | 2.42 [2.25, 2.59] | 2.78 [2.63, 2.93] |
| **$\delta D$ (‰)** | -384 [-397, -372] | -315 [-326, -303] | -251 [-257, -246] |



**Table 3.** Distributions of deseasonalized daily mean IWV and $\delta D_{col}$ data (FTIR Zugspitze) for STI and TNA events identified from Garmisch lidar measurements in 2013–2015 given as mean ± 2 SE (for STI also mean ± 1 SE is given).

| | Mean ± 2 SE | | Mean ± 1 SE | |
| --- | --- | --- | --- | --- |
| | IWV (mm) | $\delta D_{col}$ (‰) | IWV (mm) | $\delta D_{col}$ (‰) |
| **STI** | 4.18 [3.85, 4.51] | -322 [-333, -310] | 4.18 [4.01, 4.34] | -322 [-327, -316] |
| **All w/o STI** | 4.61 [4.29, 4.94] | -302 [-312, -292] | 4.61 [4.45, 4.77] | -302 [-307, -297] |
| **All** | 4.41 [4.18, 4.64] | -311 [-319, -304] | 4.41 [4.29, 4.52] | -311 [-315, -307] |
| **All w/o TNA** | 4.30 [4.05, 4.54] | -316 [-324, -308] | | |
| **TNA** | 5.47 [4.90, 6.05] | -266 [-284, -247] | | |





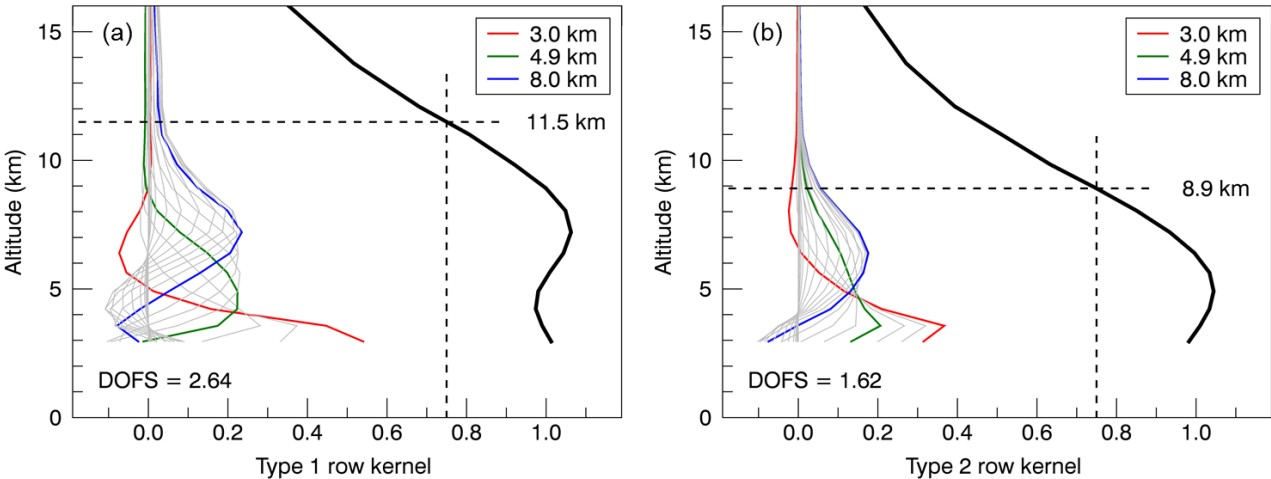

**Figure 1.** Averaging kernel rows of an exemplary $H_2O$ measurement at Zugspitze (29 October 2009, 13:21 UTC) for two retrieval products: (a) optimally estimated humidity state (type 1) and (b) consistent $\{H_2O, \delta D\}$ state (type 2). Thick black lines show the vertical column sensitivity (sum of row elements of the averaging kernel matrix).





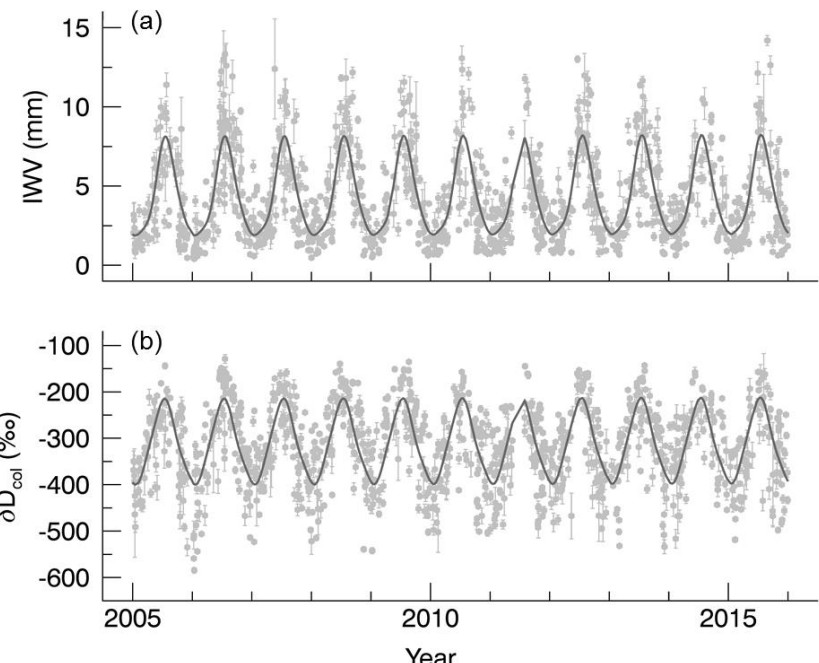

**Figure 2.** Daily mean time series of (a) integrated water vapor and (b) column-based $\delta$D retrieved from Zugspitze FTIR measurements. Error bars indicate uncertainties of daily means ($\pm$ 2 SE) and grey lines show corresponding seasonal cycles (fitted third-order Fourier series).



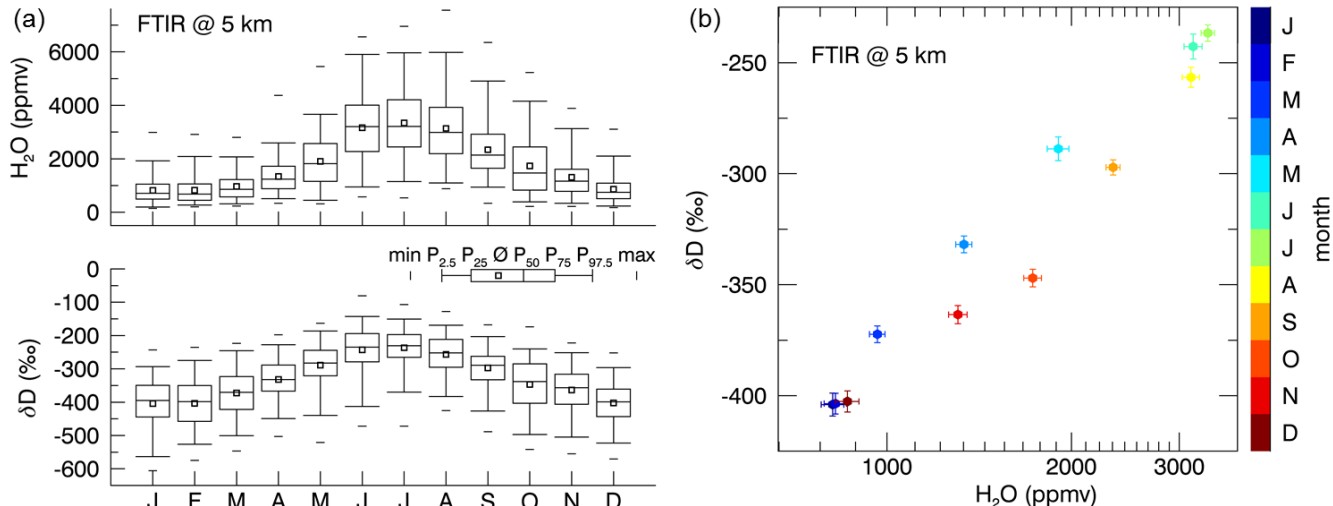

**Figure 3.** Seasonal cycles of water vapor and $\delta D$ in the free troposphere above Mt. Zugspitze: (a) multiannual monthly means and frequency distributions (percentiles P specified in the legend) of FTIR data in 5 km a.s.l. and (b) {$H_2O$, $\delta D$} plot for multiannual monthly means with error bars indicating standard errors of monthly means ($\pm$ 2 SE).




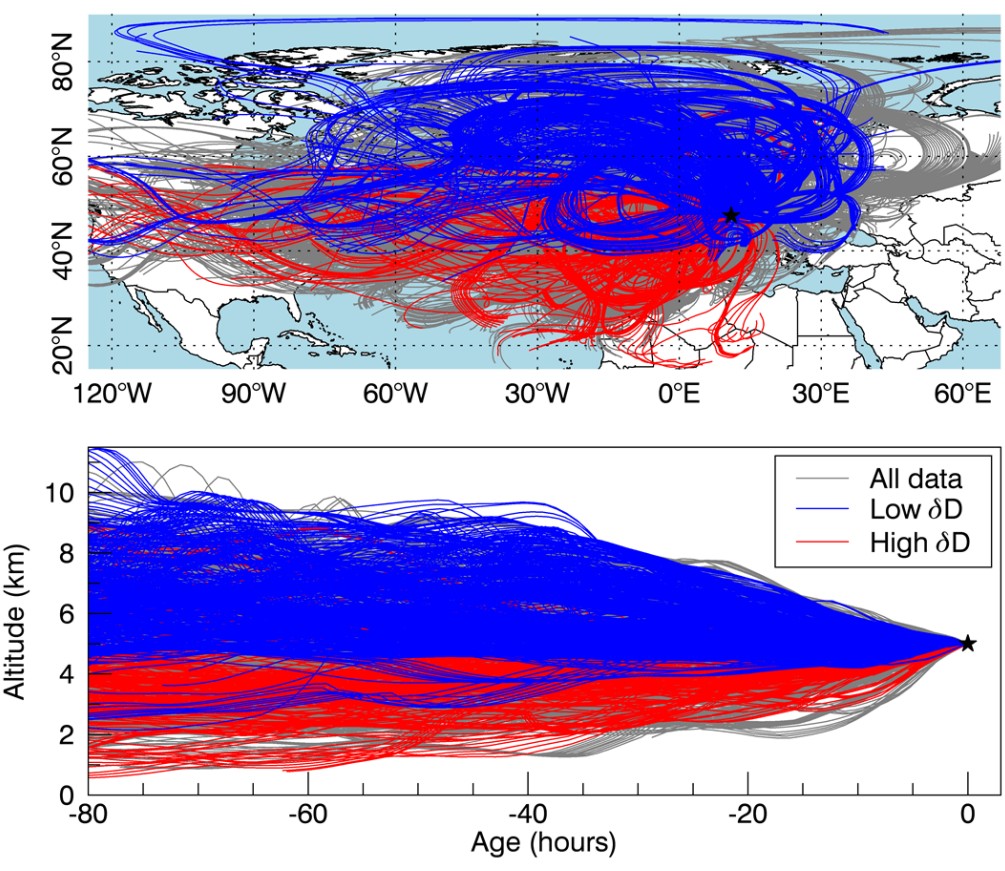

**Figure 4.** Backward trajectories of air masses arriving at Mt. Zugspitze (marked as black star) in 5 km a.s.l. altitude: map projection (upper panel) and vertical cross section (lower panel) of trajectories for all FTIR measurement times in 2005–2015 (grey lines) and for days identified as $\delta D_{col}$ outliers (blue and red lines).



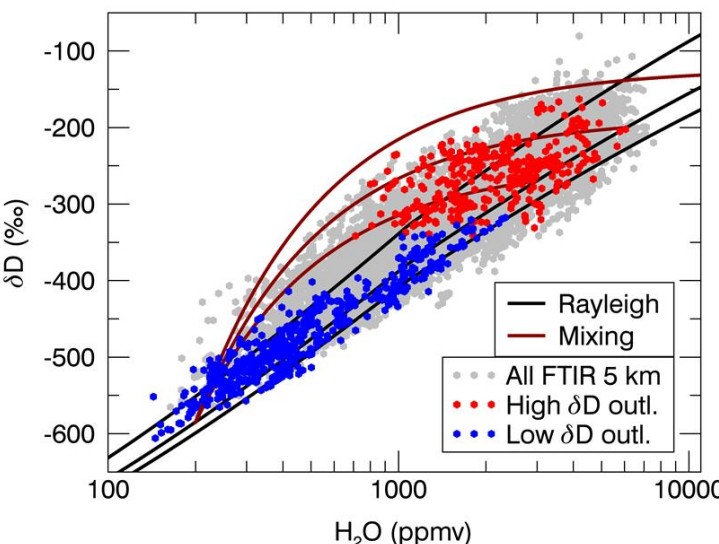

**Figure 5.** Free-tropospheric measurements of {$H_2O$, $\delta D$} pairs above Zugspitze (5 km a.s.l.) for all 2005–2015 data and for high and low $\delta D$ outliers as determined from the deseasonalized column-based $\delta D$ time series. Simulated Rayleigh and mixing processes are shown for comparison.





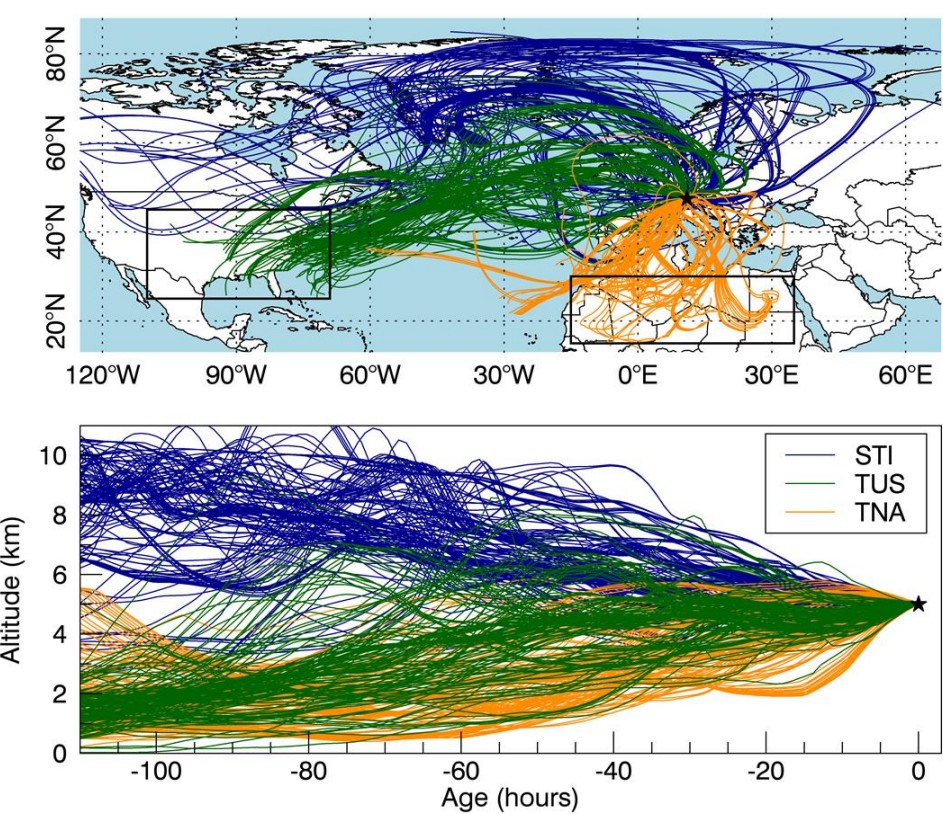

**Figure 6.** Categorization of backward trajectories for long-range transport events: map projection (upper panel) and vertical cross section (lower panel) for stratospheric intrusions (STI), transport from North America (TUS), and from Northern Africa (TNA). Black boxes indicate source regions used to classify TUS and TNA (see Table 2) and black star marks Mt. Zugspitze.





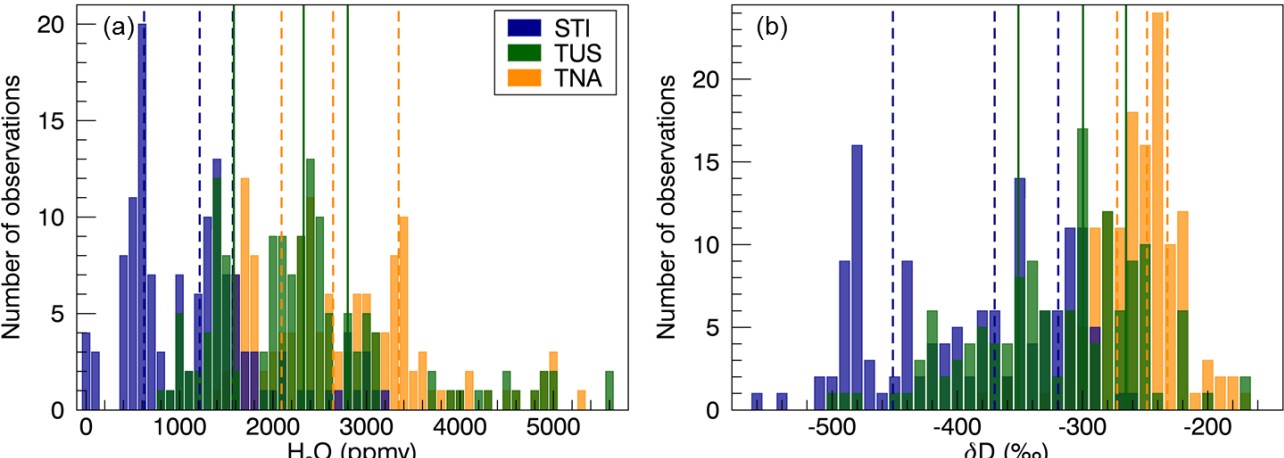

**Figure 7.** {$H_2O$, $\delta D$} signatures of long-range transport events: distributions of deseasonalized (a) $VMR_{H2O}$ and (b) $\delta D$ in the free troposphere above Zugspitze (5 km a.s.l.) for stratospheric intrusions (STI), transport from North America (TUS), and from Northern Africa (TNA) as identified in Fig. 6. Vertical lines show median, 25th and 75th percentiles of the distributions.





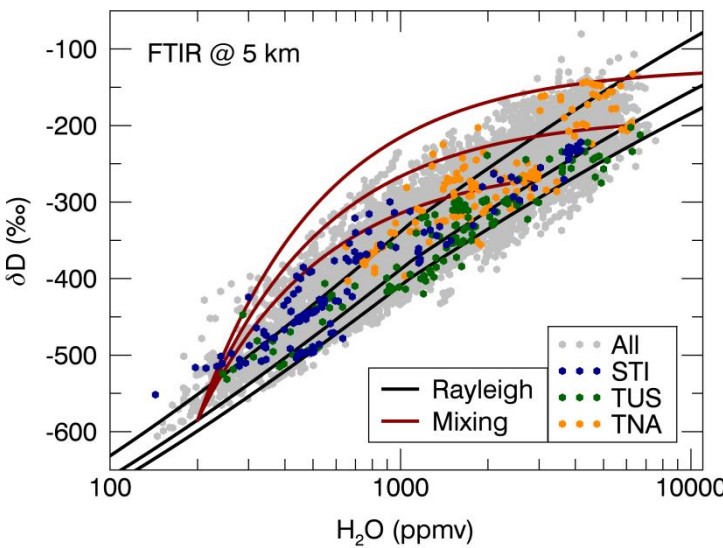

**Figure 8.** Free-tropospheric measurements of {$H_2O$, $\delta D$} pairs above Zugspitze (5 km a.s.l.) for all 2005–2015 data and for long-range transport events of stratospheric intrusions (STI), transport from North America (TUS), and from Northern Africa (TNA). Simulated Rayleigh and mixing processes are shown for comparison (compare Fig. 5).





**Figure 9.** Distributions of deseasonalized daily mean IWV and column-based $\delta$D for days identified as (a, b) stratospheric intrusion events (STI) and as (c, d) dust transport events from North Africa (TNA). For comparison distributions for all FTIR measurement days in 2013–2015 are shown as well as distributions excluding the respective transport events. Vertical lines indicate median and 25$^{th}$–75$^{th}$ percentile range for each distribution (STI / all without STI and TNA / all without TNA).





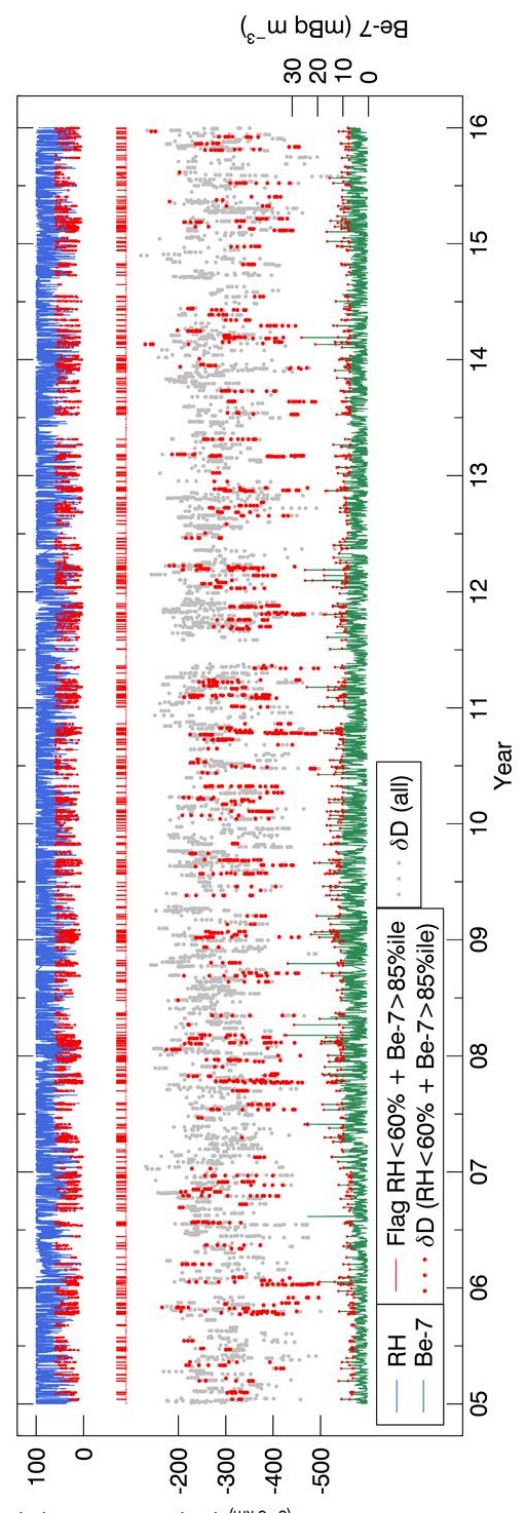

**Figure 10.** Deep stratospheric intrusions at Mt. Zugspitze in 2005–2015 derived from in situ observations of stratospheric tracers (relative humidity and Be-7) using DSTI-flag 1 and coincident FTIR data of lower-tropospheric δD (deseasonalized hourly means of 3-5 km partial columns).

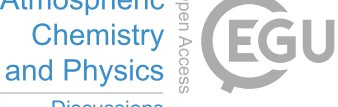

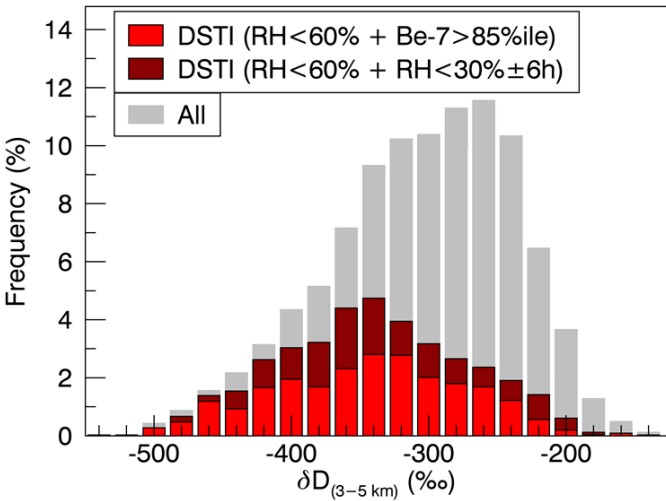

**Figure 11.** Distribution of lower-tropospheric $\delta$D (deseasonalized hourly means) for deep stratospheric intrusion events identified from Zugspitze in situ observations using two different flags (DSTI-flag 1: RH < 60 % and Be-7 > 85[th] percentile; DSTI-flag 2: RH < 60 % and 1 × RH < 30 % within ± 6h). For comparison the distribution is shown for all FTIR measurements coincident with in situ data in 2005–2015 as well as all data excluding DSTI events. Vertical lines indicate median and 25[th]–75[th] percentile range of the distribution for DSTI-flag 1 and all data without DSTI-flag 1.