# Peer review of "A decadal time series of water vapor and D/H isotope ratios above Mt. Zugspitze: transport patterns to Central Europe"

_Atmospheric Chemistry and Physics, 2016_

## Referee Comment (RC1) · Anonymous Referee #1 · 25 Feb 2017

General Comments

This manuscript presents a decade of water vapor and $\delta$D measurements above Mt. Zugspitze retrieved from mid-infrared FTIR solar absorption spectra. Several data products are retrieved: water vapor profiles with DOFS = 2.8, {H2O, $\delta$D} pairs with DOFS = 1.6 and maximum vertical sensitivity at $\sim$5 km, integrated water vapor (IWV), and integrated $\delta$D column. The weakly positive IWV trend is found to be statistically insignificant, although consistent with the 2005-2015 temperature increase observed at Mt. Zugspitze, assuming constant relative humidity. Strong seasonal cycles are observed in IWV and the $\delta$D column, with summer maxima and winter minima that are attributed to changes in the sea surface temperature of the Atlantic Ocean.

Large ensembles of HYSPLIT back-trajectories are used to identify atmospheric transport pathways to the Central European free troposphere. Low $\delta$D columns are found to be associated with the descent of dry air masses from higher latitudes, while high high $\delta$D columns are associated with the ascent of moist air masses from lower latitudes. The trajectory analysis is used to show that {H2O, $\delta$D} pairs have distinct signatures that can be related to (i) intercontinental transport from North America (TUS), (ii) intercontinental transport from North Africa (TNA), and (iii) stratospheric air intrusions (STI). For TUS events, {H2O, $\delta$D} values depend on surface temperature in the source region and the degree of dehydration during updraft in warm conveyor belts. TNA events involve dry convection of boundary layer air, resulting in relatively moist and weakly HDO-depleted air masses, while STI events bring predominantly dry and HDO-depleted air masses.

In situ measurements at Mt. Zugspitze and lidar profiles measured at Garmisch are used to verify TNA and STI transport events from 2013–2015. This work demonstrates that {H2O, $\delta$D} observations at Mt. Zugspitze can be used as a proxy for the transport of trace gases to the Central European free troposphere, e.g., pollution from North America, mineral dust from North Africa, and ozone from the stratosphere. This represents an interesting application of this relatively new dataset.

The manuscript is well written and provides a clear description of the work. I recommend publication in ACP after the minor comments below are addressed.

Specific Comments

Page 1, lines 11-12 and page 6, para 2 – Although the trend in IWV is described as statistically insignificant, it is still related to the positive trend in temperature. Is this valid? What is the correlation between IWV and temperature? Could add a third panel to Figure 2 showing the temperature time series.

Page 5, line 6 – Define LMDZ and give some information about the model a priori profiles used – which constituents? mean of profiles over some time period? specifically

for Mt. Zugspitze or mean over some region? etc.

Page 6, line 31 – It is not obvious from panel b of Figure 3a that the dD monthly frequency distributions are moderately left-skewed, which is used to conclude that there is episodic influence by strongly HDO-depleted air masses. Can this be better illustrated or justified?

Page 7, line 20 – Explain what "Based on positive experience" means.

Page 10, lines 20-22 – Why is a climatological tropopause altitude used to identify STI events? Justify this choice.

Page 11, lines 3-6 – This paragraph repeats much of the last paragraph of Section 4.1 (page 8, lines 32-35). Revise to reduce repetition, or move both to the Conclusions section.

Page 12, lines 10-15 – Some additional information could be provided to explain and justify the tracer thresholds used identify stratospheric intrusions.

Page 13, line 2 – R = -0.295 means R-squared = 0.09, although 99% is given as the confidence. Although statistical analysis allows a relationship to be weak but significant, R-squared = 0.09 indicates a very weak correlation between dD and Be-7. How useful is this?

In general – State clearly what the uncertainty bounds are. These seem to vary throughout the manuscript and are defined in some cases and not others, making it unclear what is meant in some cases. Consistency would be helpful. For example: - page 1, line 24 says "uncertainty of $\pm$ 2 standard errors" (can the standard error be asymmetric?), - page 4, line 8 says "uncertainty given as 95 % confidence interval", - page 11, line 31 says "significance on 2-$\sigma$ level", - page 12, line 4 says "mean $\pm$ 1 SE" - page 12, line 24 says "(mean $\pm$ 2 SE)"

Technical Corrections

Page 1, lines 24-25 – Give the altitude represented by the VMR values.

Page 2, line 1 – delete "above"

Page 2, line 2 – delete "potential" (if the $H_2O$, dD observations are being used as a proxy, then the transport has occurred)

Page 2, line 2 – change to "database"

Page 2, line 12 – "regional climate AND air quality, as well as . . ."

Page 2, line 19 – "THE hydrological cycle"

Page 2, line 23 – "Along these major transport pathways, other trace gas signatures and pollution plumes also can travel over . . ."

Page 2, line 24 – why "– possibly even to other continents" when this is known to occur? delete this phrase

Page 3, line 1 – "such as [Mt.?] Zugspitze AND Jungfraujoch"

Page 3, line 3 – ". . . modeling HAVE PREVIOUSLY identified"

Page 3, line 8 – "typically after descending"

Page 3, line 21 – ". . . sets have become available"

Page 3, line 30 – ". . . sensing, a promising new"

Page 3, line 31 – delete "vast"

Page 3, line 32 – "associated WITH the"

Page 4, line 4 – "representative OF"

Page 4, line 6 – add comma after "dataset"

Page 4, line 7 – "and TO combine"

Page 4, line 19 – add comma after "NDACC"

Page 5, line 13 – "as THE sum"

Page 5, line 13 – Is "exemplary" the correct word here? Exemplary means perfect or the best. Perhaps "typical" is more appropriate?

Page 5, line 18 – "sensitivity AT 5 km"

Page 5, line 22 – "root-mean-square (RMS) residuals of the spectral fit, which"

Page 5, line 28 – Are these "quality-selected spectra" the result of applying the RMS residual threshold described in the previous paragraph? If so, for clarity, could say "10184 FTIR spectra selected after filtering by the RMS residual as described above, with . . ."

Page 6, line 1 – "site)." delete extra period

Page 6, line 3 – "at THE nearby"

Page 6, line 30 – add comma after "measurements"

Page 7, line 15 – "than THE 95th"

Page 7, line 18 – "measurements, 120-hour"

Page 7, line 21 – define NCEP

Page 7, line 21 – "estimated to BE 10-20 %"

Page 7, line 23 – Clarify whether the initial point is in time (i.e., the end point of the back-trajectory) or in space (the first point of the back-trajectory). Also, explain why the point of last condensation is chosen as the initial point.

Page 7, line 24 – "defined as THE region"

Page 7, line 25 – "120-hour"

[Figure]

Page 7, line 26 – "in THE absence"

Page 7, line 30 – "at THE nearby"

Page 8, line 2 – add comma after "observations"

Page 8, line 6 – "only if AN LC point"

Page 8, line 32 – "The dD-outlier analysis reveals the potential of . . ."

Page 8, line 34 – "as A useful"

Page 9, lines 2-3 – "observations also provide information on long-range"

Page 9, line 17 – "reach THE Northern"

Page 9, lines 20 and 25– suggest breaking up this paragraph, starting new paragraphs at "The second category" and at "The third transport class"

Page 9, line 22 – add comma after "Each year"

Page 9, line 23 – add comma after "Europe"

Page 9, line 30 – here and elsewhere, Zugspitze or Mt. Zugspitze?

Page 9, line 31 – define ECMWF

Page 10, line 3 – change "above" to "at higher altitudes"

Page 10, line 32 – "In THE case"

Page 11, line 11 – "In THE case"

Page 11, line 4 – delete "valuable" – not necessary

Page 11, line 16 – "at THE Garmisch site"

Page 11, line 25 – "4-day" "315-h"

Page 11, line 31 – add comma after "transport"

Page 12, line 11 – add comma after "unambiguous"

Page 13, line 31 – "implies that weakly depleted air masses are also found"

Page 13, line 12 – "Both IWV and dDcol exhibit"

Page 13, line 16 – delete "vast"

Page 13, line 22 – don't need to redefine STI, TUS, and TNA

Page 13, line 27 – change "from" to "using"

Page 13, line 28 – "nearby"

Page 13, line 29 – "on the 2-sigma"

Page 13, line 30 – change "of" to "using"

Page 14, line 1 – "confirm the importance"

Page 14, line 8 – delete "valuable"

Page 14, line 25 – change "on the coupling" to "of the coupling"

Page 14, line 31 – "We thank the D. . . and . . . for support."

Page 22, Table 1 caption – "and THE range"

Page 22, Table 1, last row – give the altitude (range) for the H2O VMR

Page 25, Figure 1 caption – Is "typical" really meant, rather than "exemplary" (i.e., best)?

Page 26, Figure 2 – could add a third panel with the temperature time series

Page 27, Figure 3 caption – "data at 5 km a.s.l."

Page 28, Figure 4 caption – "at 5 km a.s.l."

Page 29, Figure 5 caption – More information is needed to describe the three curves

each shown for the Rayleigh and mixing processes.

Figures 6-9 – No need to redefine STI, TUS, and TNA in all four captions. Already defined in the main text.

Page 33, Figure 9 caption – add comma after "For comparison"

Page 34, Figure 10 – This figure could be improved for clarity. The labelling and grouping of the legends could be improved, e.g., better contrast between blue and green lines in the legend, group RH terms together and dD terms together. What are the red bars going across the plot between RH at the top and dD at the bottom?

---

## Referee Comment (RC2) · Anonymous Referee #2 · 31 Mar 2017

General comments

This paper presents results of recently produced H2O and $\delta$D FTIR measurements from the Mt. Zugspitze observatory. It uses an extensive set of backward trajectory analysis runs to show there are distinct transport patterns through which moisture reaches the site: intercontinental transport from North America (TUS) and North Africa (TNA), as well as stratospheric air intrusions (STI). Lidar profile measurements verify TNA and STI events. This study shows Mt. Zugspitze can monitor long-range transport events potentially impacting the Central European free troposphere and stratospheric ozone, and demonstrates the value of using {H2O, $\delta$D} pair analysis as a tracer for transport processes.

I recommend the manuscript be published in ACP.

I have a few minor comments related to the clarity of the text for the authors' consideration:

Specific minor comments

Throughout the text, the authors use both "Mt. Zugspitze" and "Zugspitze" interchangeably to refer to the observatory site. I suggest remaining consistent, and to consider using the latter for the observatory site exclusively (or an alternative) because "Mt. Zugspitze" is needed in the manuscript as a reference to the geographic landform, e.g. page 12 line 9.

Page 5 line 13: The authors say they've selected an "exemplary" measurement to represent the mean state of the Zugspitze time series. What criteria were applied to justify "exemplary"?

Page 7 line 20: I suggest the authors clarify what is meant by "based on positive experience". This is unclear.

Page 7, line 25: The decision to use 120 hour trajectories if no condensation point is found aligns with other studies; however, I suggest adding a clarification about why this duration is appropriate.

Page 8, line 2 and 3: I suggest giving the reader threshold values for what is considered "extraordinarily low" and "very high" $\delta$Dcol in the text.

Page 9, line 25 (and elsewhere): STI already defined. Re-check the manuscript and remove extra definitions.

Page 13, line 2: The reported R value of -0.295 indicates a very weak correlation. I feel it is necessary for the authors to discuss this result and justify the asserted relationship between $\delta$D and Be-7.

Page 29, Figure 4: It is regrettable that the backward trajectories overwhelm the figure's

geographical information. Interpretation of the results would benefit if the figure could retain the outline of the continents. Most of the map is not visible.

---

## Author Comment (AC1) · 8 May 2017

**Author's response on behalf of all co-authors**

Petra Hausmann, 8 May 2017

Karlsruhe Institute of Technology, Garmisch-Partenkirchen, Germany

We would like to thank both anonymous referees for their efforts and their constructive comments which helped us to improve our manuscript. Please find below our point by point response to each comment.

**1    Response to Referee #1**

**SPECIFIC COMMENTS**

**Page 1, lines 11-12 and page 6, para 2** – Although the trend in IWV is described as statistically insignificant, it is still related to the positive trend in temperature. Is this valid? What is the correlation between IWV and temperature? Could add a third panel to Figure 2 showing the temperature time series.

> *We agree that – in general – no sound conclusion should be drawn from statistically insignificant trend estimates. Nevertheless, we would like to point out that the most probable IWV trend derived would be consistent with the temperature increase in consideration of its confidence interval. Furthermore, we would like to note that adding temperature data to Fig. 2 would not provide relevant information beyond the temperature trend communicated. However, the temperature time series for Zugspitze (source: Deutscher Wetterdienst; station 5792) is publicly available at ftp://ftp-cdc.dwd.de/pub/CDC/observations_germany/climate/hourly. Concerning the correlation between IWV and temperature, we argue as follows: The correlation of simultaneous in-situ measurements of temperature and water vapor mixing ratios is defined by the Clausius–Clapeyron equation. Compared with surface mixing ratios, IWV data will exhibit lower variability. In terms of long-term trend estimates, however, IWV and surface mixing ratios are assumed to show a comparable evolution as water vapor columns are dominated by the lowest layers of the water vapor vertical profile.*

> *To take these considerations into account, we adapted the manuscript (page 1, lines 11-12) as follows: "The integrated water vapor (IWV) trend of 2.4 [-5.8, 10.6] % decade$^{-1}$ is statistically insignificant (95 % confidence interval). Under this caveat, the IWV trend estimate is conditionally consistent with the 2005–2015 temperature increase at Zugspitze (1.3 [0.5, 2.1] K decade$^{-1}$) assuming constant relative humidity". Additionally,*

*the wording on page 6 (para 2) was changed: "Over the same time period, a significant positive temperature trend of 1.3 [0.5, 2.1] K per decade is observed at Zugspitze (derived from daily means of in situ temperature measurements coincident with FTIR spectra within a period of ± 30 min; source: Deutscher Wetterdienst, available at [ftp://ftp-cdc.dwd.de/pub/CDC/observations_germany/climate/hourly](ftp://ftp-cdc.dwd.de/pub/CDC/observations_germany/climate/hourly)). [...] This IWV trend deduced from the temperature increase in consideration of its confidence interval would be formally consistent with the most probable IWV trend observed.".*

**Page 5, line 6** – Define LMDZ and give some information about the model a priori profiles used – which constituents? Mean of profiles over some time period? Specifically for Mt. Zugspitze or mean over some region? etc.

*We added more detail on the a priori profiles used: "... and a priori profiles from LMDZ-iso simulations (isotopic version of the model Laboratoire de Météorologie Dynamique-Zoom; Risi et al., 2012; globally and annually averaged $H_2O$ and HDO profiles)".*

**Page 6, line 31** – It is not obvious from panel b of Figure 3a that the dD monthly frequency distributions are moderately left-skewed, which is used to conclude that there is episodic influence by strongly HDO-depleted air masses. Can this be better illustrated or justified?

*We agree with the referee that the skewness of the $\delta D$ distributions (lower panel of Fig. 3a) is not as obvious as the positive (right-) skewness of the $H_2O$ data (upper panel of Fig. 3a). Nevertheless, several months show a clearly negative (left-) skewness, e.g. January, May and June. The other monthly distributions are either weakly left-skewed or symmetric. In order to take account of this context, we adjusted the wording in the respective sentence: "For free-tropospheric $\delta D$ measurements, monthly frequency distributions show a tendency to being moderately left-skewed (e.g., January, May, and June), which might point to episodic influence by strongly HDO-depleted air masses (e.g., originating in the upper troposphere or lower stratosphere)."*

**Page 7, line 20** – Explain what "Based on positive experience" means.

*This phrase was intended to state that the meteorological data chosen were successfully applied in previous transport studies. We reformulated the wording as follows: "We apply meteorological data from the NCEP reanalysis (global 2.5° grid), which have yielded excellent results in interpreting transport related features in lidar data for many years (e.g., Trickl et al., 2010)."*

**Page 10, lines 20-22** – Why is a climatological tropopause altitude used to identify STI events? Justify this choice.

*In this paragraph, we discuss the relatively large scatter for the $\delta D$ distribution of STI events and identified the choice of a climatological tropopause as one contributing factor – among other reasons. Details on the criteria defined for STI identification are given in the previous paragraph (page 9, lines 30-33). The choice of a dynamical*

*tropopause using potential vorticity would be more accurate, but potential vorticity data are not provided along HYSPLIT trajectories. Considering the uncertainty of HYSPLIT trajectories (approx. 10–20 % of the travel distance; Stohl, 1998), the choice of a climatological tropopause altitude occurs to be a reasonable first approximation.*

**Page 11, lines 3-6** – This paragraph repeats much of the last paragraph of Section 4.1 (page 8, lines 32-35). Revise to reduce repetition, or move both to the Conclusions section.

*We revised this paragraph to reduce repetition. The new wording is as follows: "Overall, we find distinct {$H_2O$, $\delta D$} fingerprints in Zugspitze FTIR data for three categories of long-range transport patterns (TUS, TNA, and STI). The analysis presented shows that consistent {$H_2O$, $\delta D$} observations are applicable for studying atmospheric transport events to the Central European free troposphere."*

**Page 12, lines 10-15** – Some additional information could be provided to explain and justify the tracer thresholds used to identify stratospheric intrusions.

*We expanded this paragraph with additional information on the trace thresholds applied as follows: "In a study on deep stratospheric intrusions over Central Europe, Trickl et al. (2010) examined criteria for the detection of stratospheric air intrusions based on filtering Zugspitze in situ measurements. In the following, these filtering criteria are applied to identify deep stratospheric intrusions to the summit of Mt. Zugspitze. The first indicator (flag 1) combines the occurrence of dry air masses (RH < 60 %) with simultaneous high Be-7 concentrations (Be-7 larger than the 85$^{th}$ percentile of its annual distribution). The second indicator (flag 2) identifies DSTI events by means of the same relative humidity threshold (RH < 60 %) in combination with the occurrence of very dry air masses (RH < 30 %) within six hours before or after the respective measurement. As elaborated in Trickl et al. (2010), these filtering criteria rather reliably verify stratospheric air intrusions predicted by trajectory calculations based on operational ECMWF forecasts."*

**Page 13, line 2** – R = -0.295 means R-squared = 0.09, although 99% is given as the confidence. Although statistical analysis allows a relationship to be weak but significant, R-squared = 0.09 indicates a very weak correlation between dD and Be-7. How useful is this?

*We agree that the correlation presented is very weak. This analysis was shown in order to illustrate the negative correlation expected. From our perspective, the main reason why data do not clearly meet this expectation is the relatively low vertical resolution of FTIR measurements compared to in situ data. The lower-tropospheric partial column (3–5 km a.s.l.) considered for the correlation analysis does not cover exactly the same air mass as the in situ measurements, but typically includes layers or mixtures of air masses transported from different regions. This fact is especially relevant for the analysis of stratospheric intrusions which can be very thin atmospheric filaments. Taking these considerations into account, we adapted the wording in the respective paragraph: "Correlation analysis for all coincident measurements of lower-*

*tropospheric δD (FTIR) and Be-7 (in situ) yields a significant negative correlation (99 % confidence). Although this correlation is very weak (correlation coefficient of R = -0.295), it is in line with the negative correlation expected, as Be-7 concentrations increase with altitude towards the lower stratosphere, whereas δD values decrease with altitude. The low correlation coefficient can most plausibly be explained by the different vertical sensitivity of in situ data compared to the lower-tropospheric FTIR partial columns potentially influenced by various different air masses and mixing."*

**In general** – State clearly what the uncertainty bounds are. These seem to vary throughout the manuscript and are defined in some cases and not others, making it unclear what is meant in some cases. Consistency would be helpful. For example: - page 1, line 24 says "uncertainty of ± 2 standard errors" (can the standard error be asymmetric?), - page 4, line 8 says "uncertainty given as 95 % confidence interval", - page 11, line 31 says "significance on 2-σ level", - page 12, line 4 says "mean ± 1 SE" - page 12, line 24 says "(mean ± 2 SE)"

> *Throughout the manuscript, uncertainties of mean values are given as range of ± 2 SE (standard error of the mean). This uncertainty interval is symmetric around the mean, but rounding to few decimal places can lead to an asymmetric appearance. Trend estimates are generally given with a 95% confidence interval corresponding to ± 2 standard deviations. The only case deviating from this overall definition occurs in Sect. 4.3. Here, the uncertainty is additionally given as interval of ± 1 SE, as differences observed are not significant if considering uncertainty of ± 2 SE. For reasons of consistency, we replaced the phrase "significance on 2-σ level" by "considering uncertainty of ± 2 SE" and, analogously, "significance on 1-σ level" by "considering uncertainty of ± 1 SE". In addition, we inserted an overall uncertainty definition (p. 6, line 1): "Throughout this manuscript, uncertainties of mean values are given as range of ± 2 SE, i.e., standard error of the mean $SE = SD/\sqrt{n}$ with sample size n (except for the discussion in Sect. 4.3, where also ranges of ± 1 SE are considered)".*

**TECHNICAL CORRECTIONS**

**Page 1, lines 24-25** – Give the altitude represented by the VMR values.

> *We inserted the altitude represented by the data (5 km a.s.l.) in the description.*

**Page 2, line 1** – delete "above"

> *The manuscript was changed as suggested.*

**Page 2, line 2** – delete "potential" (if the H2O, dD observations are being used as a proxy, then the transport has occurred).

> *We applied the suggested correction.*

**Page 2, line 2** – change to "database"

*The manuscript was changed as suggested.*

**Page 2, line 12** – "regional climate AND air quality, as well as ..."

*We applied the suggested correction.*

**Page 2, line 19** – "THE hydrological cycle"

*The manuscript was changed as suggested.*

**Page 2, line 23** – "Along these major transport pathways, other trace gas signatures and pollution plumes also can travel over …"

*We applied the suggested correction.*

**Page 2, line 24** – why "– possibly even to other continents" when this is known to occur? Delete this phrase

*The manuscript was changed as suggested.*

**Page 3, line 1** – "such as [Mt.?] Zugspitze AND Jungfraujoch"

*The manuscript was changed as suggested.*

**Page 3, line 3** – "… modeling HAVE PREVIOUSLY identified"

*We applied the suggested correction.*

**Page 3, line 8** – "typically after descending"

*The manuscript was changed as suggested.*

**Page 3, line 21** – "… sets have become available"

*The manuscript was changed as suggested.*

**Page 3, line 30** – "… sensing, a promising new"

*We applied the suggested correction.*

**Page 3, line 31** – delete "vast"

*The manuscript was changed as suggested.*

**Page 3, line 32** – "associated WITH the"

*The manuscript was changed as suggested.*

**Page 4, line 4** – "representative OF"

*We applied the suggested correction.*

**Page 4, line 6** – add comma after "dataset"

*The manuscript was changed as suggested.*

**Page 4, line 7** – "and TO combine"

*The manuscript was changed as suggested.*

**Page 4, line 19** – add comma after "NDACC"

*We applied the suggested correction.*

**Page 5, line 13** – "as THE sum"

*The manuscript was changed as suggested.*

**Page 5, line 13** – Is "exemplary" the correct word here? Exemplary means perfect or the best. Perhaps "typical" is more appropriate?

*As proposed, we replaced "exemplary" by "typical".*

**Page 5, line 18** – "sensitivity AT 5 km"

*The manuscript was changed as suggested.*

**Page 5, line 22** – "root-mean-square (RMS) residuals of the spectral fit, which"

*The manuscript was changed as suggested.*

**Page 5, line 28** – Are these "quality-selected spectra" the result of applying the RMS residual threshold described in the previous paragraph? If so, for clarity, could say "10184 FTIR spectra selected after filtering by the RMS residual as described above, with …"

*Yes, "quality selected spectra" refers to the criteria of quality selection described in the previous paragraph, i.e., filtering by the RMS residual and the total DOFS value. For clarification of this context, we changed the wording as follows: "… derived from 10184 FTIR spectra after filtering by quality selection criteria described above, with…".*

**Page 6, line 1** – "site)." delete extra period

*The extra period and the bracket were deleted.*

**Page 6, line 3** – "at THE nearby"

*We applied the suggested correction.*

**Page 6, line 30** – add comma after "measurements"

*The manuscript was changed as suggested.*

**Page 7, line 15** – "than THE 95th"

*The manuscript was changed as suggested.*

**Page 7, line 18** – "measurements, 120-hour"

*The manuscript was changed as suggested.*

**Page 7, line 21** – define NCEP

*We added the following definition: "…NCEP (National Center for Environmental Prediction) reanalysis…".*

**Page 7, line 21** – "estimated to BE 10-20 %"

*The manuscript was changed as suggested.*

**Page 7, line 23** – Clarify whether the initial point is in time (i.e., the end point of the back-trajectory) or in space (the first point of the back-trajectory). Also, explain why the point of last condensation is chosen as the initial point.

> *We specified the meaning of "initial point" and added an explanation on choosing the point of last condensation as initial point. The revised version of this paragraph is formulated as follows: "The initial point of each trajectory (i.e., the location from which the air mass was transported to Mt. Zugspitze) is chosen as the point of last condensation (LC), defined as the region where the relative humidity along the trajectory exceeds 80 % over a three-hour period (see González et al., 2016). Conditions at the point of last condensation determine the air parcel's water vapor mixing ratio and its isotopic composition, as these quantities are conserved in the absence of sources and sinks (Galewsky et al., 2005; Noone et al., 2012), i.e., if mixing during transport is negligible and no further condensation occurred. If no LC point exists along the trajectory, full 120-hour trajectories are depicted."*

**Page 7, line 24** – "defined as THE region"

> *We applied the suggested correction.*

**Page 7, line 25** – "120-hour"

> *The manuscript was changed as suggested.*

**Page 7, line 26** – "in THE absence"

> *We applied the suggested correction.*

**Page 7, line 30** – "at THE nearby"

> *The manuscript was changed as suggested.*

**Page 8, line 2** – add comma after "observations"

> *The manuscript was changed as suggested.*

**Page 8, line 6** – "only if AN LC point"

> *We applied the suggested correction.*

**Page 8, line 32** – "The dD-outlier analysis reveals the potential of …"

> *The manuscript was changed as suggested.*

**Page 8, line 34** – "as a useful"

> *We applied the suggested correction.*

**Page 9, lines 2-3** – "observations also provide information on long-range"

> *The manuscript was changed as suggested.*

**Page 9, line 17** – "reach THE Northern"

*We applied the suggested correction.*

**Page 9, lines 20 and 25**– suggest breaking up this paragraph, starting new paragraphs at "The second category" and at "The third transport class"

*Extra line breaks were included to split this paragraph.*

**Page 9, line 22** – add comma after "Each year"

*We applied the suggested correction.*

**Page 9, line 23** – add comma after "Europe"

*We applied the suggested correction.*

**Page 9, line 30** – here and elsewhere, Zugspitze or Mt. Zugspitze?

*Following the suggestion by referee #2, we applied a consistent wording throughout the manuscript, i.e., Mt. Zugspitze for the geographical location and Zugspitze for the observatory site.*

**Page 9, line 31** – define ECMWF

*The following definition was added and moved to a separate sentence: "Zonal mean tropopause altitudes were taken from ECMWF data (European Centre for Medium-Range Weather Forecasts) as given in Eckhardt et al. (2004)".*

**Page 10, line 3** – change "above" to "at higher altitudes"

*We applied the suggested correction.*

**Page 10, line 32** – "In THE case"

*We applied the suggested correction.*

**Page 11, line 1** – "In THE case"

*The manuscript was changed as suggested.*

**Page 11, line 4** – delete "valuable" – not necessary

*We applied the suggested correction.*

**Page 11, line 16** – "at THE Garmisch site"

*We applied the suggested correction.*

**Page 11, line 25** – "4-day" "315-h"

*The manuscript was changed as suggested.*

**Page 11, line 31** – add comma after "transport"

*We applied the suggested correction.*

**Page 12, line 11** – add comma after "unambiguous"

*We applied the suggested correction.*

**Page 12, line 31** – "implies that weakly depleted air masses are also found"

*The manuscript was changed as suggested.*

**Page 13, line 12** – "Both IWV and dDcol exhibit"

*We applied the suggested correction.*

**Page 13, line 16** – delete "vast"

*We applied the suggested correction.*

**Page 13, line 22** – don't need to redefine STI, TUS, and TNA

*The manuscript was changed as suggested.*

**Page 13, line 27** – change "from" to "using"

*We applied the suggested correction.*

**Page 13, line 28** – "nearby"

*We applied the suggested correction.*

**Page 13, line 29** – "on the 2-sigma"

*This phrase was deleted from the manuscript in response to the Referee's comment on a consistent definition of uncertainties.*

**Page 13, line 30** – change "of" to "using"

*The manuscript was changed as suggested.*

**Page 14, line 1** – "confirm the importance"

*We added "the" to this phrase.*

**Page 14, line 8** – delete "valuable"

*The manuscript was changed as suggested.*

**Page 14, line 25** – change "on the coupling" to "of the coupling"

*The manuscript was changed as suggested.*

**Page 14, line 31** – "We thank the D... and … for support."

*We applied the suggested correction.*

**Page 22, Table 1 caption** – "and THE range"

*We applied the suggested correction.*

**Page 22, Table 1, last row** – give the altitude (range) for the H2O VMR

*All data included in Table 1 reflect average conditions at the last condensation points identified for trajectories on δD outlier days. This also applies to the VMR values in the last row – the respective altitude range can be found in the second row.*

**Page 25, Figure 1 caption** – Is "typical" really meant, rather than "exemplary" (i.e., best)?

*We intended to say "typical" and applied the suggested correction.*

**Page 26, Figure 2** – could add a third panel with the temperature time series

*As stated in the response to the first Referee's comment, we would like to note that adding temperature data to Fig. 2 would not provide relevant additional information and the temperature time series for Zugspitze is publicly available (source: Deutscher Wetterdienst).*

**Page 27, Figure 3 caption** – "data at 5 km a.s.l."

*We applied the suggested correction.*

**Page 28, Figure 4 caption** – "at 5 km a.s.l."

*We applied the suggested correction.*

**Page 29, Figure 5 caption** – More information is needed to describe the three curves each shown for the Rayleigh and mixing processes.

*More detailed information was added in the caption as follows: "For comparison, simulated Rayleigh processes (initial evaporation at T = 15 °C, RH = 80%, δD values of -60 ‰, -130 ‰, and -160 ‰) as well as mixing processes of upper-tropospheric air ($VMR_{H2O}$ = 200 ppmv, δD = -585 ‰) with three lower-middle-tropospheric air masses (i. $VMR_{H2O}$ = 13500 ppmv, δD = -130 ‰; ii. $VMR_{H2O}$ = 6100 ppmv, δD = -200 ‰; iii. $VMR_{H2O}$ = 3000 ppmv, δD = -270 ‰.) are shown."*

**Figures 6-9** – No need to redefine STI, TUS, and TNA in all four captions. Already defined in the main text.

*These redundant definitions have been removed from the captions.*

**Page 33, Figure 9 caption** – add comma after "For comparison"

*We applied the suggested correction.*

**Page 34, Figure 10** – This figure could be improved for clarity. The labelling and grouping of the legends could be improved, e.g., better contrast between blue and green lines in the legend, group RH terms together and dD terms together. What are the red bars going across the plot between RH at the top and dD at the bottom?

*We improved this figure for clarity. In the new version, green and blue graphs can be better distinguished and the legend was simplified. The red bars mentioned by the referee indicated the DSTI flag as true/false info. We removed this graph from the revised figure as this information is redundant as already visualized by the red marks on the other graphs.*

**2 Response to Referee #2**

**Throughout the text,** the authors use both "Mt. Zugspitze" and "Zugspitze" interchangeably to refer to the observatory site. I suggest remaining consistent, and to consider using the latter for the observatory site exclusively (or an alternative) because "Mt. Zugspitze" is needed in the manuscript as a reference to the geographic landform, e.g. page 12 line 9.

> *We thank the referee for pointing out this inconsistency and applied the suggested wording throughout the manuscript, i.e., Mt. Zugspitze for the geographical location and Zugspitze for the observatory site.*

**Page 5, line 13:** The authors say they've selected an "exemplary" measurement to represent the mean state of the Zugspitze time series. What criteria were applied to justify "exemplary"?

> *As stated by Referee #1, "typical" would be the more appropriate wording. We selected a typical measurement representing the mean state of Zugspitze time series with respect to IWV and DOFS of the type 2 product.*

**Page 7, line 20:** I suggest the authors clarify what is meant by "based on positive experience". This is unclear.

> *As stated in response to a similar comment of Referee #1, this phrase was reformulated as follows: "We apply meteorological data from the NCEP reanalysis (global 2.5° grid), which have yielded excellent results in interpreting transport related features in lidar data for many years (e.g., Trickl et al., 2010)."*

**Page 7, line 25:** The decision to use 120 hour trajectories if no condensation point is found aligns with other studies; however, I suggest adding a clarification about why this duration is appropriate.

> *As the referee states, 120-hour trajectories have been chosen in several previous transport studies (e.g., González et al., 2016; Škerlak et al., 2014; Trickl et al., 2014). The choice of five-day backward trajectories is considered to be appropriate in relation to typical transport time scales. North American emissions typically reach the free troposphere above Europe within 4–6 days (Stohl et al., 2002). Mineral dust from the Saharan region is usually transported to Europe within 2–5 days (Papayannis et al., 2008). Direct stratospheric intrusions to the Northern Alps typically travel 2–4 days from the lower stratosphere (Trickl et al., 2010; Škerlak et al., 2014). Furthermore, the increase in trajectory uncertainty with increasing trajectory duration (i.e., travel distance) has to be taken into account as well as increasing influence of mixing and exchange processes. Additionally, the choice of five-day trajectories is consistent with the lifetime of species most relevant for intercontinental transport (e.g., about one week for ozone or aerosols; Holloway et al., 2003). In this context, we added the following clarification to the manuscript: "The choice of five-day trajectories appears appropriate in consideration of typical transport time scales, lifetimes of transported species, and trajectory uncertainty."*

**Page 8, line 2 and 3:** I suggest giving the reader threshold values for what is considered "extraordinarily low" and "very high" $\delta$Dcol in the text.

> *With these terms the authors refer to the outlier definition given on page 7, line 15-16. For reasons of clarity, we repeated the threshold values applied at this point.*

**Page 9, line 25 (and elsewhere):** STI already defined. Re-check the manuscript and remove extra definitions.

> *Redundant definitions of STI, TNA, and TUS were removed from the manuscript.*

**Page 13, line 2:** The reported R value of -0.295 indicates a very weak correlation. I feel it is necessary for the authors to discuss this result and justify the asserted relationship between $\delta$D and Be-7.

> *As stated in response to an analogous comment by Referee #1 related to this correlation analysis, we agree that the derived correlation is very weak. For discussion of this result, we adapted the paragraph as follows: "Correlation analysis for all coincident measurements of lower-tropospheric δD (FTIR) and Be-7 (in situ) yields a significant negative correlation (99 % confidence). Although this correlation is very weak (correlation coefficient of R = -0.295), it is in line with the negative correlation expected, as Be-7 concentrations increase with altitude towards the lower stratosphere, whereas δD values decrease with altitude. The low correlation coefficient can most plausibly be explained by the different vertical sensitivity of in situ data compared to the lower-tropospheric FTIR partial columns potentially influenced by various different air masses and mixing."*

**Page 29, Figure 4:** It is regrettable that the backward trajectories overwhelm the figure's geographical information. Interpretation of the results would benefit if the figure could retain the outline of the continents. Most of the map is not visible.

> *We adapted the figure in order to retain the outline of the continents.*

**3   Further changes proposed by the authors**

**Page 15, line 4:** Update reference with final revised version.

**Page 35, Figure 11:** Unfortunately, an old figure version was included in the discussion manuscript (compare newer version of Fig. 11 in the manuscript uploaded for registration). We inserted this newer version which is consistent with the existing text and figure caption.

[revised manuscript text omitted]

---

## Author Response (AR1)

Dear Dr Peuch,

please find our point-by-point response to the reviews together with a marked-up manuscript version including all changes at http://www.atmos-chem-phys-discuss.net/acp-2016-1029/acp-2016-1029-AC1-supplement.pdf.

We think the discussion phase was very fruitful as we were able to positively reply to all referee comments and perform related improvements to the manuscript.

We hope our paper is herewith mature to be accepted for final publication.
Thank you for your efforts.

Sincerely,
Petra Hausmann, Karlsruhe Institute of Technology
Garmisch-Partenkirchen, 8 May 2017